# A class-specific effect of dysmyelination on the excitability of hippocampal interneurons

Delphine Pinatel[1], Edouard Pearlstein[1], Giulia Bonetto[1†], Laurence Goutebroze[2], Domna Karagogeos[3], Valérie Crepel[1], Catherine Faivre-Sarrailh[1]*

[1]INMED, INSERM, Aix Marseille Université, Marseille, France; [2]INSERM, Institut du Fer à Moulin, Sorbonne Université, Faculté des Sciences et Ingénierie, Paris, France; [3]Department of Basic Sciences, University of Crete Medical School and Institute of Molecular Biology and Biotechnology, Foundation for Research and Technology, University of Crete, Heraklion, Greece

*For correspondence:
catherine.sarrailh@univ-amu.fr

Present address: †AstraZeneca, Discovery Sciences, Cellular Assay Development Team, Cambridge, United Kingdom

**Abstract** The role of myelination for axonal conduction is well-established in projection neurons but little is known about its significance in GABAergic interneurons. Myelination is discontinuous along interneuron axons and the mechanisms controlling myelin patterning and segregation of ion channels at the nodes of Ranvier have not been elucidated. Protein 4.1B is implicated in the organization of the nodes of Ranvier as a linker between paranodal and juxtaparanodal membrane proteins to the spectrin cytoskeleton. In the present study, 4.1B KO mice are used as a genetic model to analyze the functional role of myelin in Lhx6-positive parvalbumin (PV) and somatostatin (SST) neurons, two major classes of GABAergic neurons in the hippocampus. We show that 4.1B-deficiency induces disruption of juxtaparanodal K⁺ channel clustering and mislocalization of nodal or heminodal Na⁺ channels. Strikingly, 4.1B-deficiency causes loss of myelin in GABAergic axons in the hippocampus. In particular, stratum oriens SST cells display severe axonal dysmyelination and a reduced excitability. This reduced excitability is associated with a decrease in occurrence probability of small amplitude synaptic inhibitory events on pyramidal cells. In contrast, stratum pyramidale fast-spiking PV cells do not appear affected. In conclusion, our results indicate a class-specific effect of dysmyelination on the excitability of hippocampal interneurons associated with a functional alteration of inhibitory drive.

## Editor's evaluation

This important study identifies the functional consequence of myelination of interneuronal axons on circuit function by showing that 4.1B deletion leads to altered myelination in a subset of interneurons and altered intrinsic and synaptic physiological parameters. The authors' conclusions about how myelination of inhibitory axons affects physiological properties are based on solid evidence using a combination of imaging and electrophysiological approaches.

## Introduction

Recent reports have established that subtypes of GABAergic neurons could be myelinated. Indeed, a substantial fraction of myelin, both in mouse and human neocortex, belongs to inhibitory neurons by comparison with pyramidal cells (*Micheva et al., 2016*; *Stedehouder et al., 2017*). Long-range projecting hippocampal GABAergic neurons are myelinated and play a role in the synchronization among distant brain areas (*Jinno et al., 2007*; *Melzer et al., 2012*). Surprisingly, locally projecting

interneurons such as the parvalbumin (PV) cells are frequently myelinated, independently of their morphological subtypes (i.e. basket, axo-axonic, or bi-stratified) in the mouse hippocampus. A fraction of somatostatin (SST) interneurons, which include both long-range and Oriens-Lacunosum Moleculare (O-LM) or bistratified local projecting neurons has been also reported to be myelinated (*Micheva et al., 2016*; *Stedehouder et al., 2017*; *Stedehouder et al., 2019*). Myelin optimizes action potential (AP) propagation by forming electrical insulation and restricting ion channel distribution, but its significance is still elusive in local projecting interneurons. Oligodendrocytes wrapping of GABAergic axons may be essential for providing axonal metabolic support (*Krasnow and Attwell, 2016*; *Philips and Rothstein, 2017*). Recent reports indicate that myelination of PV interneurons is modulated by neuronal activity or sensory experience and is able to shape the inhibitory circuitry (*Benamer et al., 2020a*; *Stedehouder et al., 2018*; *Yang et al., 2020*). However, how myelination could modulate the function of the different subtypes of interneurons remains poorly evaluated.

In the neocortex and hippocampus, GABAergic interneurons present the particularity to be covered by myelin sheaths that display a patchy distribution along the axon (*Benamer et al., 2020b*; *Micheva et al., 2016*; *Stedehouder et al., 2017*).The discontinuous myelination of inhibitory ramified axons implies the presence of heminodal structures along mature axons. The mechanisms regulating myelin coverage as well as segregation of ion channels at heminodes and nodes of Ranvier remain to be elucidated. Remarkably, hippocampal PV and SST neurons display clusters of $Na^+$ channels forming prenodes along the axons before myelination (*Bonetto et al., 2019*; *Freeman et al., 2016*). These prenodes can be induced by oligodendrocyte-secreted matrix molecules including Contactin (*Dubessy et al., 2019*). Such $Na^+$ channel clusters along pre-myelinated inhibitory axons are associated with increased conduction velocity (*Freeman et al., 2015*). We also showed that the axons of hippocampal PV and SST interneurons are highly enriched before myelination in Kv1 channels that may regulate firing during development. The Kv1 channels are associated with the cell adhesion molecules Contactin2/TAG-1, Caspr2, and ADAM22 and the scaffolding protein 4.1B forming complexes in the juxtaparanodal domains of the nodes of Ranvier after myelination (*Bonetto et al., 2019*). It is however so far unknown whether deficiency in any of the juxtaparanodal components Contactin2/TAG-1 encoded by *Cntn2*, Caspr2 encoded by *Cntnap2* or 4.1B encoded by *Eph41l3* could affect the myelination of hippocampal interneurons. Here, we focused our study on the consequences of 4.1B-deficiency.

Protein 4.1B belongs to a superfamily of proteins that share a conserved FERM domain (4.1/ezrin/radixin/moesin) and serve as membrane cytoskeleton linkers and participate in a wide variety of cellular events such as motility and cell adhesion (*Denisenko-Nehrbass et al., 2003*). In myelinated axons, protein 4.1B is implicated in the organization of the nodes of Ranvier as a linker between paranodal and juxtaparanodal membrane proteins to the actin/spectrin cytoskeleton. At juxtaparanodes, 4.1B is linked to the Caspr2/Contactin2 cell adhesion complex and is required for the clustering of the Kv1 channels as reported in the PNS and CNS (*Buttermore et al., 2011*; *Cifuentes-Diaz et al., 2011*; *Einheber et al., 2013*; *Horresh et al., 2010*). At paranodes, 4.1B mediates the linkage between the Caspr/Contactin adhesion complex and ßII-spectrin and participates to the boundary with the nodal complex enriched in Nav channels and Neurofascin186 anchored to ßIV-spectrin/AnkyrinG scaffold (*Brivio et al., 2017*; *Zonta et al., 2008*). In addition to its role in the organization of ion channel domains at the nodes of Ranvier, 4.1B linked with Caspr at paranodes has been also reported to promote internodal elongation during development (*Brivio et al., 2017*).

We discovered that 4.1B KO mice display selective aberrant myelination of GABAergic interneurons in the hippocampus. An intriguing question is whether hippocampal interneurons require paranodal cytoskeleton as an instructive cue regulating the extent of myelin sheath coverage. We study the cellular features of SST and PV cells in the hippocampus by examining ion channel distribution, axon initial segment (AIS) topography and electrophysiological intrinsic properties. We show that 4.1B-deficiency induces a severe and selective loss of myelin sheath along SST and PV axons in the stratum radiatum of CA1 hippocampus. Notably, the excitability of SST cells located in the stratum oriens is specifically reduced in contrast to PV cells not affected in the stratum pyramidale. This study reveals the functional role of myelin in affecting the excitability of subtypes of hippocampal interneurons.

## Results

### Deficiency in the scaffolding protein 4.1B causes severe loss of myelination in inhibitory axons of the stratum radiatum in the hippocampus

We used the transgenic *Lhx6-tdTomato* reporter line (Jackson laboratory) to fluorescently label subtypes of GABAergic neurons. Lhx6 is a transcription factor expressed in inhibitory neuron subclasses originating from the medial ganglionic eminence, which includes all hippocampal PV and SST neurons (*Liodis et al., 2007*). Double-immunostaining for MBP as a myelin marker, and Caspr as a paranodal marker was performed and we evaluated that 75 ± 5% of the myelinated axons were Lhx6-positive in the stratum radiatum at P35 (*Figure 1—figure supplement 1A, D*; *Figure 1—source data 1*).

We showed previously that cell adhesion and scaffolding molecules associated in complex with juxtaparanodal Kv1 are highly expressed in hippocampal GABAergic interneurons (*Bonetto et al., 2019*). Contactin2 is selectively expressed in SST cells in the stratum oriens and PV cells in the stratum pyramidale of the rat hippocampus (*Bonetto et al., 2019*). Here, we ask whether deficiency in any of the juxtaparanodal components Contactin2/TAG-1, Caspr2 or 4.1B could induce dysmyelination of hippocampal interneurons. Strikingly, we observed that 4.1B KO mice displayed a severe loss of myelin sheaths in the hippocampus as shown in *Figure 1*. The myelin pattern was selectively altered in the hippocampus of the 4.1B KO mice, markedly in the stratum radiatum both at P35 (*Figure 1B*) and P70 (*Figure 1F and H*, and *Figure 1—figure supplement 1H*) as an indication that myelination of inhibitory interneurons may be severely affected. Only a slight dysmyelination was observed in the molecular layer of the dentate gyrus, which receives numerous excitatory projections from the entorhinal cortex (*Figure 1B* inset and Figure 3D). We did not observe any massive loss of myelin outside the hippocampus, as shown in myelinated tracts like the corpus callosum (*Figure 1B*) or in the layers of the somato-sensory cortex (*Figure 1—figure supplement 2*). In contrast, Contactin2-deficiency did not markedly impair myelination in the CA1 hippocampus as illustrated at P70 (*Figure 1D*). In *Cntn2$^{-/-}$* mice crossed with the *Lhx6-tdTomato* reporter line, the total length of myelinated axons and the percentage of Lhx6-positive myelinated axons in the stratum radiatum were similar to control mice as quantified at P35 (*Figure 1—figure supplement 1A–D*; *Figure 1—source data 1*). As reported in several studies, *Cntnap2* KO mice show a reduced density of PV-positive interneurons in the hippocampus associated with decreased inhibitory synaptic transmission in CA1 pyramidal neurons (*Paterno et al., 2021*; *Peñagarikano et al., 2011*). However, we did not observe any apparent reduction of myelinated PV inhibitory axons in the hippocampus of *Cntnap2* KO mice at P70 (*Figure 1E* and *Figure 1—figure supplement 1E, F*).

In the hippocampus of 4.1B KO mice, the severe reduction of myelin sheaths occurred already at P25 in the stratum radiatum indicating that it originated from a developmental defect, taking place at an early stage of hippocampal myelination (*Figure 1—figure supplement 3*). Such myelin pattern alteration persisted in mature animals as shown at the age of 6 months (*Figure 1—figure supplement 4*). Since the extent of myelination was difficult to be quantified in regions with a dense and intermingled network of myelinated axons, the number of internodes was estimated from the number of paranodes immunolabeled for Caspr in the different layers of the CA1 hippocampus at P70 (*Figure 1G–I*; *Figure 1—source data 1*). The density of paranodes was significantly reduced by 85% (p=0.0060, Student's t test) in the stratum radiatum of 4.1B-deficient mice (n=4 mice/genotype). The density of paranodes tended also to decrease in the stratum oriens (–29%; p=0.1774) and the stratum pyramidale (–34%; p=0.0667). The density of paranodes was not affected in the alveus (p=0.8002) nor in the stratum lacunosum-moleculare (p=0.6273) suggesting that myelinated axons connecting extra-hippocampal areas may be preserved. In particular, this is an indication that the axons of pyramidal neurons that run in the alveus should be properly myelinated. The total length of MBP-positive axonal segments was measured at different levels of the CA1 hippocampal layers, starting from the stratum oriens (bin1) and the pyramidal layer (bin2), and throughout the stratum radiatum (bin3-7) towards the stratum lacunosum-moleculare (*Figure 1J*; *Figure 1—source data 1*). Myelination was significantly decreased by 47% (p=0.0018, Student's t test) in the stratum oriens and by 75% (p=0.0015) in the stratum radiatum of 4.1B-deficient mice, with a 69–90% reduction in the middle region of the layer (bin4: p=0.0004, bin5: p=0.0053, bin6: p=0.0113). Therefore, the fact that myelin mostly belongs to Lhx6-positive axons in the CA1 stratum radiatum, suggests that myelination of inhibitory interneurons

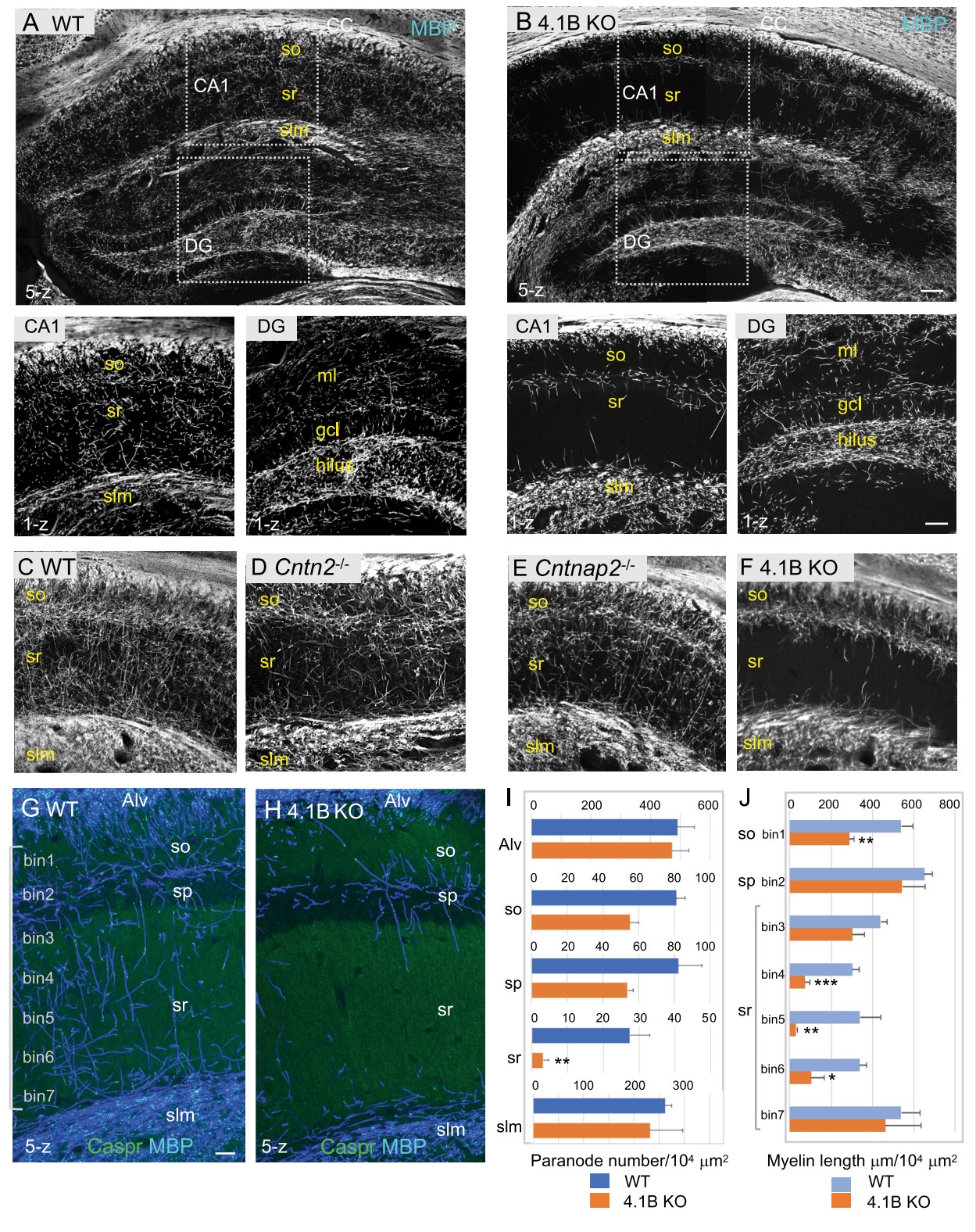

**Figure 1.** 4.1B KO mice show strong reduction of myelin in the adult hippocampus. (**A–B**) Hippocampal vibratome sections from P35 wild-type and 4.1B KO mice immunostained for MBP as a myelin marker. Note the lack of myelin sheaths in the CA1 stratum radiatum of the hippocampus whereas myelin of the corpus callosum (CC) is preserved in 4.1B KO mice. Insets show high magnification of CA1 regions and dentate gyrus (DG) (**C–F**) Hippocampus from P70 wild-type, *Cntn2-/-*, *Cntnap2-/-*, and 4.1B KO mice immunostained for MBP. Only 4.1B KO mice show a selective and massive loss of myelin in the stratum radiatum. (**G, H**) Double-staining for MBP (blue) and Caspr (green) as a paranodal marker. Maximum intensity of confocal images (5-z steps

*Figure 1 continued on next page*

*Figure 1 continued*

of 2 µm). (**I**) Quantification of the number of paranodes/$10^4$ µm$^2$ in the different layers of the CA1 hippocampus. (**J**) Quantification of the total myelin length/$10^4$ µm$^2$ in the so (bin1), sp (bin2), and sr divided in 5 bins (40x300 µm). Mean ± SEM of 4 mice for each genotype. Significant difference by comparison with wild-type: * p<0.05, ** p<0.01, and *** p<0.001 using the Student's t test. Alv: Alveus, so: stratum oriens, sp: stratum pyramidale, sr: stratum radiatum, slm: stratum lacunosum moleculare, ml: molecular layer, gcl: granule cell layer. Bar: 200 µm in (**A, B**) 50 µm in insets and (**C–F**) 25 µm in (**G, H**).

The online version of this article includes the following source data and figure supplement(s) for figure 1:

**Source data 1.** Length of myelinated axons and paranode density in the different layers of the CA1 hippocampus of wild-type and 4.1B KO mice.

**Figure supplement 1.** Myelin pattern in the CA1 hippocampus of *Cntn2*$^{-/-}$, *Cntnap2*$^{-/-}$, and 4.1B KO mice.

**Figure supplement 2.** Myelin pattern is preserved in the cortex of 4.1B KO mice at P70.

**Figure supplement 3.** Myelin pattern alteration in the hippocampus of 4.1B KO mice at P25.

**Figure supplement 4.** Myelin pattern alteration in the hippocampus of 4.1B KO mice at P180.

---

may be strongly reduced in the hippocampus of 4.1B-deficient mice. However, we evaluated the percentage of Lhx6-positive myelinated axons in the stratum radiatum of 4.1B KO mice (*Figure 1—figure supplement 1D*; *Figure 1—source data 1*), which appeared similar to control (66.1 ± 3.6% in 4.1B KO versus 75 ± 5% in control) as an indication that dysmyelination was not restricted to Lhx6 GABAergic axons.

It was first critical to evaluate whether the lack of myelin could be due to a loss of GABAergic interneurons extending their axons throughout the stratum radiatum. We first quantified the density of PV and SST cells in the stratum oriens and stratum pyramidale of CA1, which was similar in the wild-type and 4.1B KO mice (*Figure 2A–F*). Distinct interneurons subdivide their connections to sub-compartments of hippocampus pyramidal neurons with the PV basket cells connecting the soma, the bistratified cells connecting basal and apical dendrites in the stratum oriens and radiatum, and the SST O-LM cells projecting to the distal apical dendrites in the stratum lacunosum-moleculare (*Somogyi and Klausberger, 2005*) (see schematic illustration Figure 7A, H). In addition, long-range projecting SST neurons have many running axons in the stratum radiatum (*Gulyás et al., 2003*; *Jinno et al., 2007*). The 4.1B KO mice did not show a reduced density of PV cells in the stratum pyramidale (*Figure 2A and B*) as quantified at P35 in the CA1 region (*Figure 2E*; *Figure 2—source data 1*). A high density of PV axons was present in the stratum radiatum of the 4.1B KO mice (*Figure 2H*) as observed in the wild-type (*Figure 2G*), albeit they were only partly myelinated in the region bordering the stratum pyramidale (arrow in *Figure 2H*) and the total length of myelinated PV axons was significantly reduced (–63%, p=0.0317, Mann-Whitney test, n=5 ROIs, 3 mice/genotype) (*Figure 2K*; *Figure 2—source data 1*). The density of SST interneurons in the stratum oriens was also preserved (*Figure 2C, D and F*; *Figure 2—source data 1*). We observed SST-positive axons crossing the stratum radiatum in a perpendicular orientation and ramifying at the level of the stratum lacunosum-moleculare, both in the 4.1B KO mice (*Figure 2J*) as in the wild-type (*Figure 2I*). These particular SST axons are likely O-LM axons. However, these O-LM axons were only partly myelinated in the lower and upper regions of the stratum radiatum (arrows in *Figure 2J*) in the 4.1B KO mice with a myelin coverage of 42.3 ± 4.0% (*Figure 2M*; *Figure 2—source data 1*; *Video 1*) while myelin coverage was 79.7 ± 3.4% for wild-type mice (*Figure 2I and M*, *Video 2*). The total length of myelinated SST axons was significantly reduced (–58%, p<0.0001, Mann-Whitney test, n=10 ROIs, 3 mice/genotype) in the stratum radiatum (*Figure 2L*; *Figure 2—source data 1*). Thus, 4.1B-deficiency strongly reduces myelin ensheathing of PV and SST axons in the stratum radiatum.

Next, we asked whether such myelin loss could be due to a reduced number of oligodendrocytes in the hippocampus using immunostaining for Olig2 as a marker for the oligodendrocyte lineage (*Figure 3A–D*). The density of Olig2-positive cells in the stratum radiatum was 22.3±1.6/$10^5$ µm$^2$ in the 4.1B KO mice versus 25.3±2.2/$10^5$ µm$^2$ in wild-type animals at P35 (*Figure 3E*; *Figure 3—source data 1*). We did not observe any decrease in the number of Olig2-positive cells in the CA1 layers that could explain the massive loss of myelin in 4.1B KO mice. We also examined the density of Olig2-positive cells in the dentate gyrus since the neuronal progenitors of the granule cell layer might contribute to the generation of oligodendrocytes in demyelinating pathological conditions such as in the cuprizone toxic model (*Jessberger et al., 2008*; *Klein et al., 2020*). The pattern of MBP-positive axons was apparently slightly disorganized in the molecular layer and not in the hilus of the dentate gyrus in 4.1B

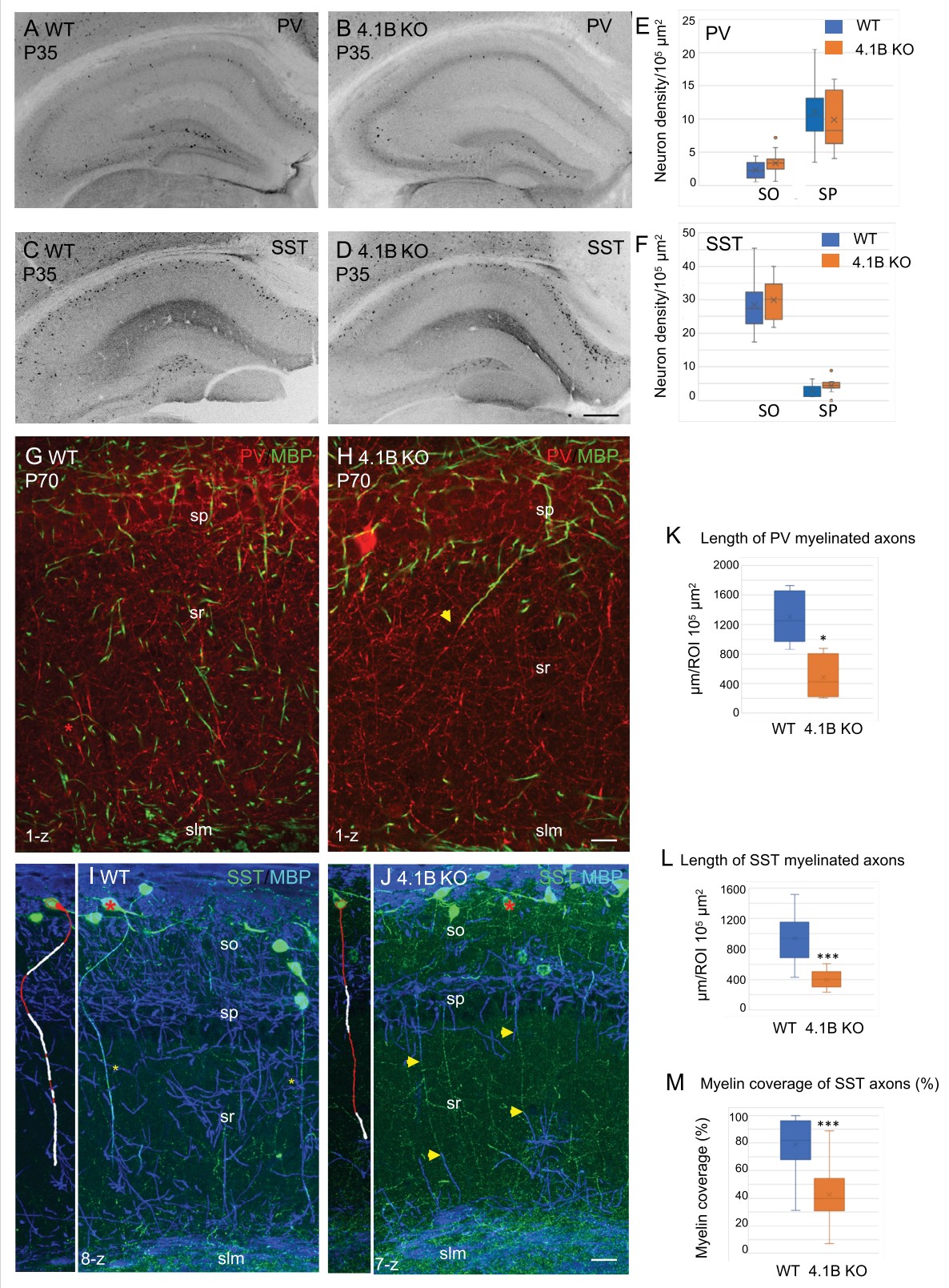

**Figure 2.** Parvalbumin and somatostatin axons are dysmyelinated in the 4.1B KO hippocampus. (**A–D**) Hippocampal sections of wild-type and 4.1B KO mice immunostained for PV (**A, B**) or SST (**C, D**) at P35. (**E, F**) Quantification of the cell density/$10^5$ µm$^2$ in the CA1 stratum oriens (SO) and stratum pyramidale (SP). No statistical difference in cell density between the genotypes (PV cell density: SO, p=0.108; SP, p=0.412 and SST cell density: SO, p=0.493; SP, p=0.057; Student's t test; n=10–12 ROIs in hippocampal sections at 2 levels in the antero-posterior axis, 3 mice/genotype). (**G, H**) Double-

*Figure 2 continued on next page*

*Figure 2 continued*

staining for PV (red) and MBP (green) at P70 showing that PV axons are present in the stratum radiatum and poorly myelinated in the 4.1B KO mice (**H**). The arrow in H points to a PV axon, which is partly myelinated. (**K**) Quantification of the total length of PV myelinated axon/$10^5$ μm$^2$ in the stratum radiatum at P70 (n=5 ROIs, 3 mice/genotype). (**I, J**) Double-staining for SST (green) and MBP (blue) at P35 showing O-LM SST cells in the stratum oriens extending their axon through the stratum radiatum to project into the stratum lacunosum moleculare. Insets show 3D reconstructions of myelinated O-LM neurons indicated with red asterisks (myelin in white). Note that SST axons in the stratum radiatum are fully myelinated in the WT (asterisks in I) and partly myelinated in 4.1B KO mice (arrows in J). (**L, M**) Quantitative analyses of the total length of SST myelinated axon/$10^5$ μm$^2$ in the stratum radiatum at P35 (n=10 ROIs, 3 mice/genotype) and the percentage of myelin coverage of individual SST axons (n=29–35 axons, 3 mice/genotype; the mean length of selected axons is 174±9 μm in WT and 197±12 μm in 4.1B KO mice). Significant difference by comparison with wild-type: * p<0.05; *** p<0.001; Mann-Whitney test. Bar: 200 μm in (**A-D**) 20 μm in (**G, H**) 25 μm in (**I, J**).

The online version of this article includes the following source data for figure 2:

**Source data 1.** Density of PV and SST cells and length of myelinated PV and SST axons in the hippocampus of wild-type and 4.1B KO mice.

KO mice at P35 (*Figure 3D*). We did not observe any difference in the density of Olig2-positive cells in the hilus, granule cell, or molecular layers of the dentate gyrus (*Figure 3F*; *Figure 3—source data 1*). These results suggest that the developmental loss of myelin induced by the deficiency in the axonal cytoskeletal linker 4.1B may not be associated with change in oligodendrogenesis.

## 4.1B-deficient GABAergic axons display reduced internodal length

Protein 4.1B mediates interactions between membrane adhesion complexes and the spectrin cytoskeleton and binds Caspr at paranodes as schematized in *Figure 4A*. We previously showed that 4.1B promotes, via its interaction with Caspr, the growth of oligodendroglial processes and myelin sheath convergence in the developing spinal cord (*Brivio et al., 2017*). Immunofluorescence staining for 4.1B in the hippocampus indicated that the scaffolding protein was detected at the soma of GABAergic PV interneurons and not at the soma of pyramidal neurons in the CA1 pyramidal layer at P70 (*Figure 4D*). As a control for specificity, no staining was observed on hippocampal sections from 4.1B KO mice (*Figure 4C*). An enrichment of Protein 4.1B in GABAergic neurons has been previously reported in premyelinated hippocampal cell culture (*Bonetto et al., 2019*). Protein 4.1B immunostaining was present in PV and SST axons along the internodes and enriched at paranodes whereas the nodal gap was unlabeled (*Figure 4E and F*). We asked whether the elongation of internodes could be affected in the 4.1B-deficient CA1 hippocampus at P70 and analyzed the distribution of the length of internodes (*Figure 4B*; *Figure 4—source data 1*). Analysis of the cumulative frequency indicates a significant difference between the genotypes (p<0.0001; Kolmogorov-Smirnov test). The mean

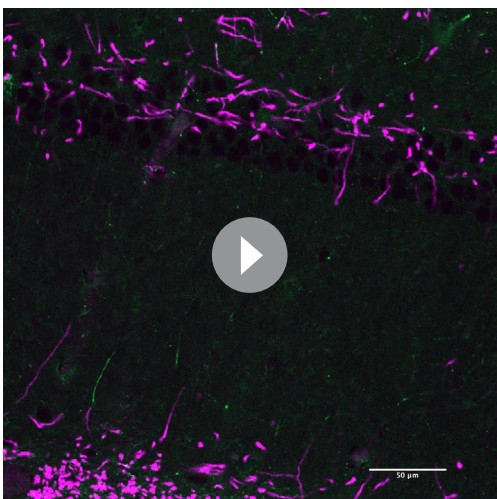

**Video 1.** O-LM axons are partly myelinated in the lower and upper regions of the stratum radiatum in 4.1B KO mice at P35. Double-staining for MBP (pink) and SST (green). 25-z steps of 1 μm.
https://elifesciences.org/articles/86469/figures#video1

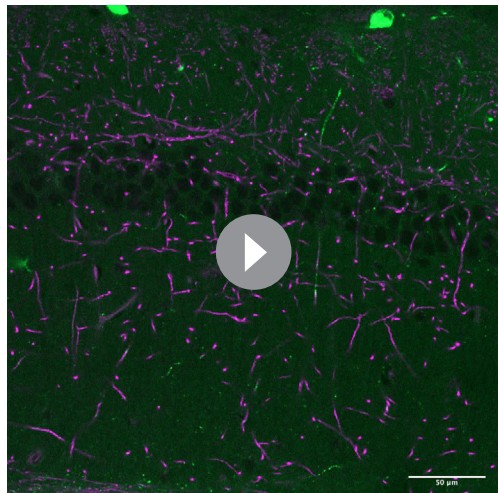

**Video 2.** O-LM axons crossing the stratum radiatum are myelinated in wild-type mice at P35. Double-staining for MBP (pink) and SST (green). 25-z steps of 1 μm.
https://elifesciences.org/articles/86469/figures#video2

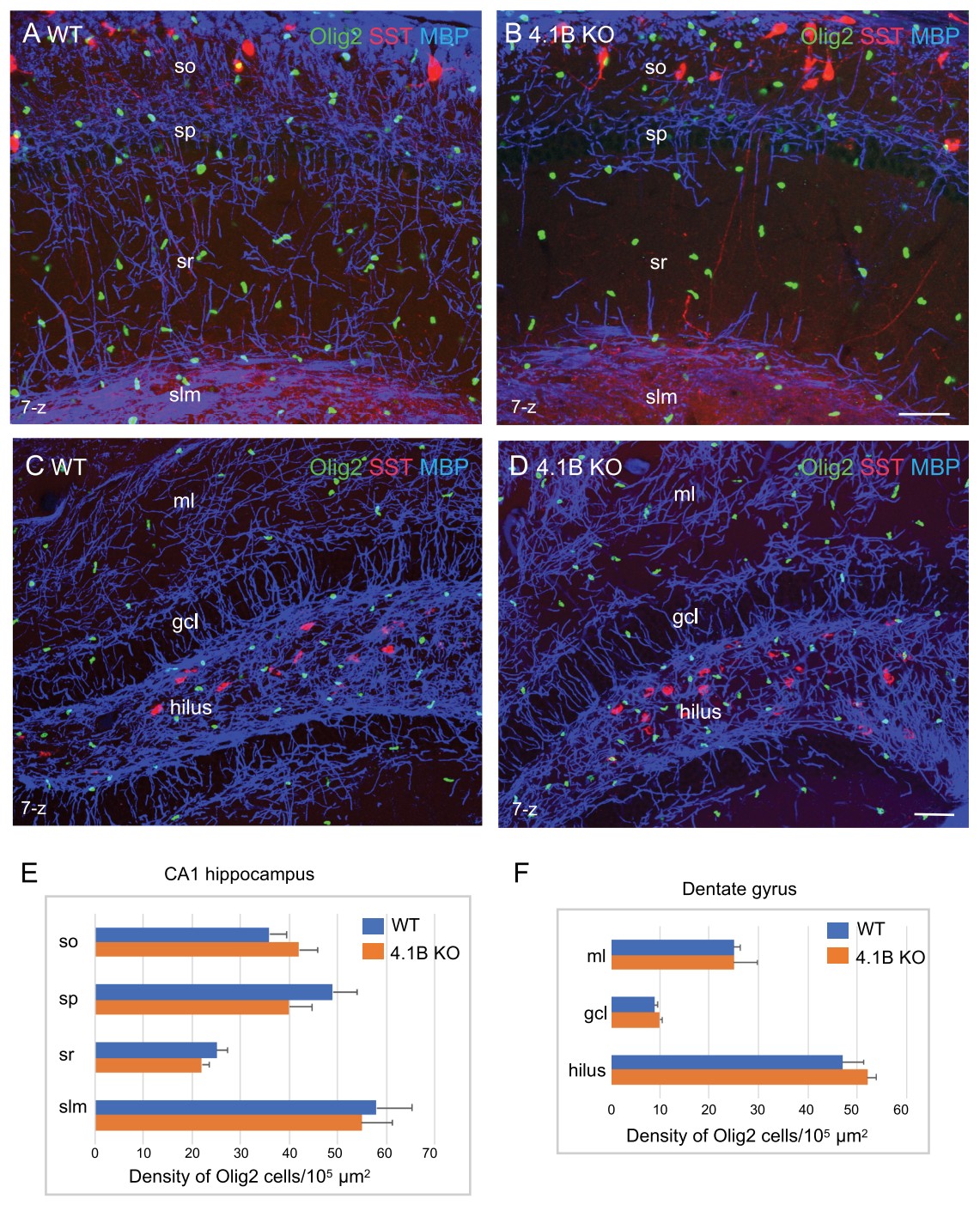

**Figure 3.** The density of oligodendroglial cells is preserved in the 4.1B KO hippocampus. Hippocampal sections of wild-type and 4.1B KO mice at P35 immunostained for MBP (blue), SST (red) and Olig2 (green) as a marker of the oligodendrocyte lineage. CA1 region (**A, B**) and dentate gyrus (**C, D**), maximum intensity of confocal images from 7-z steps of 2 μm. Bar: 50 μm. (**E, F**) Quantitative analysis of the density in Olig2-positive cells/$10^5$ μm$^2$ in the different layers of the CA1 hippocampus (n=11–12 ROIs, 3 mice/genotype) and dentate gyrus (ml: molecular layer; gcl: granular cell layer; n=3 ROIs; 3 mice/genotype). No significant difference between the genotypes (so: p=0.311; sp: p=0.171; sr: p=0.262; slm: p=0.806; ml: p=1; gcl: p=0.491; hilus: p=0.262; Student's t test).

The online version of this article includes the following source data for figure 3:

**Source data 1.** Density of Olig2-positive cells in the CA1 hippocampus and dentate gyrus of wild-type and 4.1B KO mice.

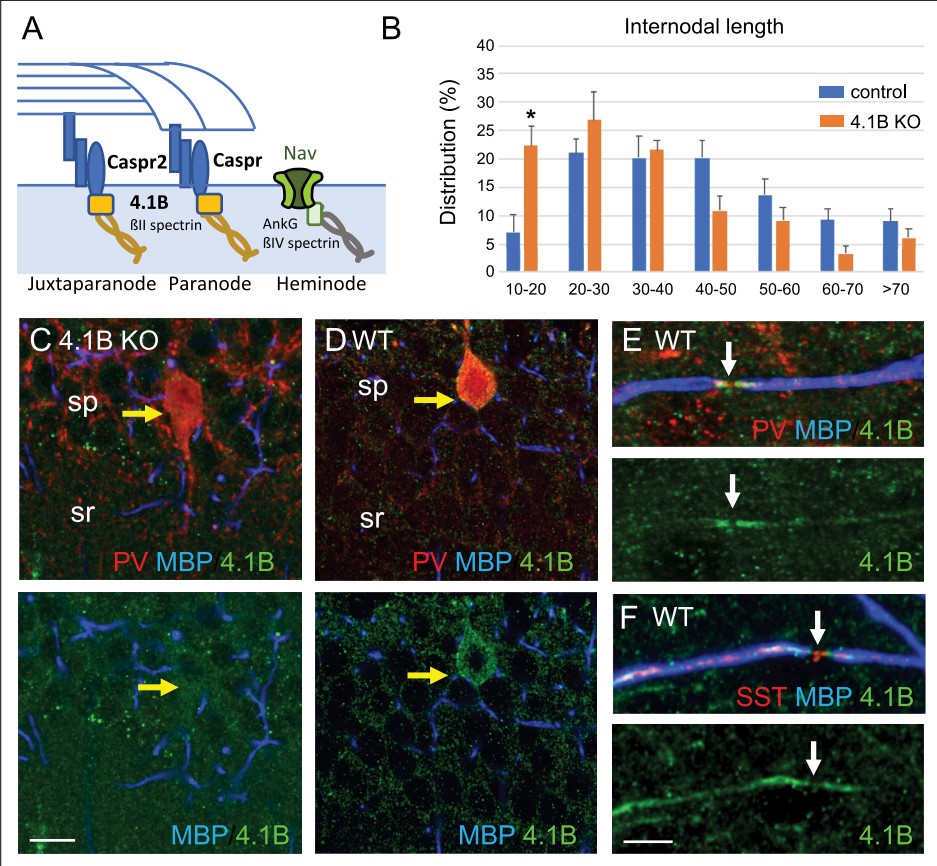

**Figure 4.** 4.1B is enriched at paranodes of GABAergic axons and involved in internodal elongation. (**A**) Schematic drawing to depict the role of 4.1B at paranodes mediating the anchorage between Caspr and the ßII-spectrin at the boundary with the nodal ßIV-spectrin. (**B**) Distribution of internodal lengths as percentage. Quantitative analysis was performed on hippocampal sections of *Lhx6-Cre;tdTomato* control and 4.1B KO mice at P70 immunostained for MBP and Caspr. Confocal imaging of 25–50 µm width z-stack were used to measure the length of myelin sheaths between two paranodes in the CA1 region (n=275 in control and n=191 in 4.1B KO mice). Significant difference by comparison with control: * p<0.05; Mann-Whitney test (Means ± SEM of 4 ROIs from 2 mice/genotype). (**C–E**) Hippocampal sections of 4.1B KO (**C**) and wild-type (**D–F**) mice at P35 immunostained for MBP (blue), PV or SST (red) and 4.1B (green). Note that 4.1B is detected at the soma of PV interneurons (**D**). As a control for specificity, no staining is detected in 4.1B-deficient hippocampus (**C**). 4.1B is enriched at paranodes and not detected at the nodal gap (arrows) in myelinated PV (**E**) and SST (**F**) axons. Bar: 10 µm in (**C, D**) 5 µm in (**E, F**).

The online version of this article includes the following source data for figure 4:

**Source data 1.** Internodal length of myelinated axons in the CA1 hippocampus of control and 4.1B KO mice.

internodal length was significantly reduced in 4.1B KO (34.5±3.1 µm; n=191) by comparison with wild-type (43.8±0.3 µm; n=275) mice (p=0.0286, Mann-Whitney test, n=4 ROIs, 2 mice/genotype). The maximum length was similarly decreased to 93.8 µm in the 4.1B KO versus 114.5 µm in the wild-type mice. The reduction of internodal length strongly indicates that protein 4.1B may be implicated in myelin sheath elongation.

## 4.1B-deficiency causes misdistribution of nodal and heminodal Nav channels in hippocampal GABAergic interneurons

At the paranodal junction, the axonal cytoskeleton including 4.1B and ßII-spectrin also organizes the boundary between the internodes and the nodal Nav channels (*Figure 4A*; *Brivio et al., 2017*; *Buttermore et al., 2011*; *Zhang et al., 2013*; *Zonta et al., 2008*). Accordingly, the Nav complex is mislocalized distant from the paranodes in the developing spinal cord of the 4.1B KO mice before the convergence of internodes (*Brivio et al., 2017*). In the adult spinal cord, this phenotype is no longer

observed and the nodal Nav clusters become properly juxtaposed to the paranodes in mature nodes. Hippocampal inhibitory axons are covered by discontinuous myelin with persistence of heminodal structures in the adult, and myelination of ramified axons implies the presence of nodes at branch points so that it was particularly interesting to analyze whether the distribution of Nav channels may be disturbed in the 4.1B-deficient axons.

We first analyzed the architecture of the nodes of Ranvier in the GABAergic interneurons of the CA1 hippocampus of wild-type mice at P30 and P70 by performing panNav channel immunostaining. As illustrated in *Figure 5A*, myelination along Lhx6-positive axons was partly discontinuous in the stratum radiatum at P70, leaving uncovered axonal segments (asterisks). Nav channels were detected at nodes (*Figure 5B, B' and G*) and branch points (*Figure 5C and C'*). Heminodal clustering of Nav channels was observed for interspaced internodes (*Figure 5E*). We noticed that the lengths of nodal Nav immunostaining measured between two paranodes were higher at P30 than at P70. Analysis of the cumulative frequency indicates that the distribution of nodal Nav immunostaining lengths was significantly different with age (p=0.0264; Kolmogorov-Smirnov test, n=40 at P30 and n=70 at P70, 2 mice/age). At the age of P30, 44% of the nodes displayed a length >2 µm versus only 22% at P70, indicating that the internodes may still elongate during the late developmental stage (*Figure 5D*; *Figure 5—source data 1*).

We then investigated whether 4.1B deficiency may induce misdistribution of nodal and heminodal Nav channels in the CA1 hippocampus at P70. In 4.1B KO mice, a number of inhibitory PV or SST axons display interrupted myelination when entering the stratum radiatum showing heminodes (*Figure 2H and J*). Interestingly, we observed at heminodes of 4.1B-deficient inhibitory axons, a gap between the site of Nav clustering and the paranode stained for Caspr (*Figure 5F* and *Figure 5—figure supplement 1B*) as compared to control mice (*Figure 5E* and *Figure 5—figure supplement 1A*). The gap could reach 10 µm and displayed a mean value of 2.25±0.51 µm significantly different from the control value (0.04±0.02 µm; p<0.0001, Mann-Whitney test, n=25 heminodes, 2 mice/genotype; *Figure 5I*; *Figure 5—source data 1*). By contrast, the length of heminodal Nav clusters was similar in control (1.96±0.29 µm) and in 4.1B KO (1.84±0.12 µm) mice (p=0.427, Mann-Whitney test; *Figure 5J*; *Figure 5—source data 1*). A gap could also be observed between nodal Nav and convergent internodes stained for MBP (*Figure 5H*) in 4.1B KO mice by comparison with control ones (p=0.0029, Mann-Whitney test, n=26 in 4.1B KO and n=87 in control, 2–3 mice/genotype; *Figure 5I*; *Figure 5—source data 1*), together with an increase in the length of nodal Nav clusters (*Figure 5J*; *Figure 5—source data 1*). The mean length of nodal Nav in 4.1B KO (1.92±0.25 µm, n=26) was significantly increased by comparison with control mice (1.59±0.14 µm, n=87; p=0.0068, Mann-Whitney test). Thus, the positioning of Nav clusters is significantly disturbed at the nodes and heminodes of dysmyelinated GABAergic interneurons of 4.1B-deficient hippocampus.

## Juxtaparanodal alterations in hippocampal interneurons of *Cntn2*, *Cntnap2*, and 4.1B KO mutant mice

Finally, we analyzed the distribution of the juxtaparanodal Kv1 channels in the stratum radiatum at P70. As expected, immunostaining for Kv1.2 was detected at the juxtaparanodes or hemi-juxtaparanodes under the compact myelin including at branch points both in Lhx6- and PV-positive axons (*Figure 6A and B*). We investigated whether the deficiency in Kv1-associated CAMs may induce alteration of the juxtaparanodes in hippocampal interneurons like it was observed in myelinated tracts of the corpus callosum, optic nerves, or spinal cord for *Cntn2* KO and *Cntnap2* KO mice (*Poliak et al., 2003*; *Savvaki et al., 2008*; *Traka et al., 2003*). As analyzed in double-blind experiments, deletion of *Cntn2* or *Cntnap2* induced a strong decrease in Kv1 channel clustering at the juxtaparanodes of PV myelinated axons in the CA1 hippocampus at P70 (*Figure 6D–F*), although a residual expression of Kv1 was still present at some juxtaparanodes in both mutants. The percentage of paranodes bordered by Kv1.2 immunostaining in PV axons was significantly reduced by 30% in *Cntn2* KO and 53% in *Cntnap2* KO mice (p<0.0001, Mann-Whitney test; *Figure 6C*; *Figure 6—source data 1*). The scaffolding protein 4.1B is known to bind Caspr2 and to be required for the proper assembly of the juxtaparanodes both in the PNS and CNS (*Buttermore et al., 2011*; *Cifuentes-Diaz et al., 2011*; *Einheber et al., 2013*). We observed a strong decrease (–61%; p<0.0001, Mann-Whitney test, n=9 ROIs, 3 mice/genotype) in the Kv1 clustering at juxtaparanodes of myelinated PV axons of 4.1B KO mice at P70 (*Figure 6C and G*; *Figure 6—source data 1*). The 4.1B KO mice crossed with the *Lhx6-Cre;tdTomato* line showed

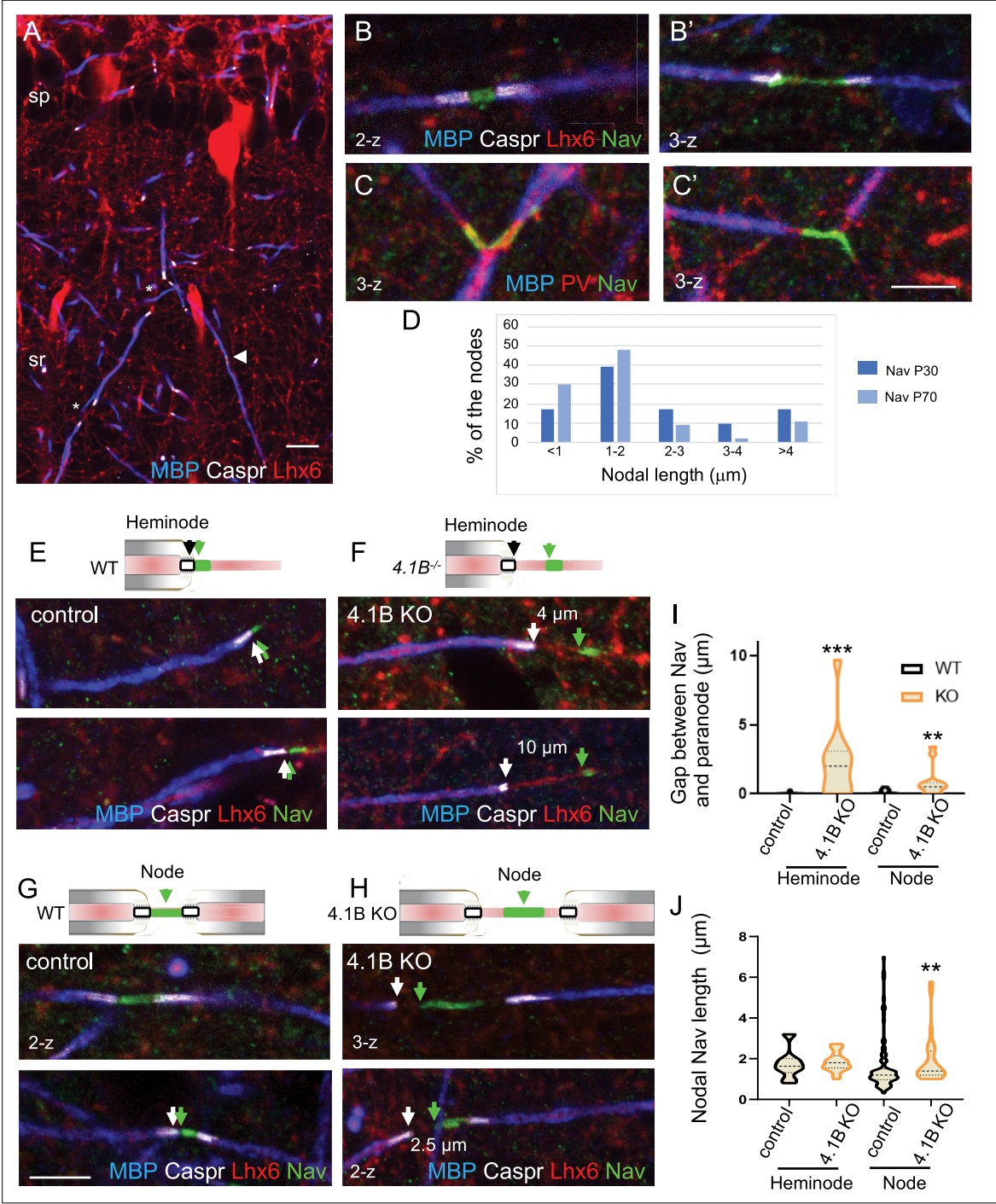

**Figure 5.** Misplacement of heminodal and nodal of Nav channels in 4.1B-deficient GABAergic axons. Hippocampal sections of P70 *Lhx6-Cre;tdTomato* control and 4.1B KO mice were immunostained for MBP (blue), Caspr (white) and panNav (green). tdTomato expressed in Lhx6-positive cells or PV immunostaining (**C**) is in red. (**A**) shows a branched myelinated Lhx6 axon with contiguous (arrowhead) and spaced (asterisks) internodes. Immunostaining for panNav at nodes with different lengths (**B, B′**) and at branch points (**C, C′**). (**D**) Distribution of nodal lengths in Lhx6-positive axons at P30 and P70 in CA1 expressed as percentages. The length of nodal Nav was measured between two paranodes (n=41 at P30 and n=64 at P70, 2 mice/ age). (**E–J**) Clustering of Nav channels at the heminodes (**E, F**) and nodes (**G, H**) of control and 4.1B KO mice in the CA1 hippocampus. Nav clusters are close to paranodes both at heminodes (**E**) and nodes (**G**) in control GABAergic axons (arrows). In contrast, a gap is frequently observed between Nav clusters and paranodes at heminodes (**F**) and nodes (**H**) in 4.1B-deficient GABAergic axons (arrows). Confocal images with maximum intensity of 2-z or 3-z steps of 1 μm. Bar: 10 μm in (**A**) 5 μm in (**B, C, E–H**). (**I**) Quantitative analysis of the distance between Nav cluster and paranode at heminodes and nodes. (**J**) Quantitative analysis of the length of nodal Nav cluster at heminodes and nodes. Significant difference by comparison with wild-type: ** p<0.01; *** p<0.001; Mann-Whitney test (heminodes: n=25/genotype; nodes: n=87 in control and n=26 in 4.1B KO mice, 2–3 mice/genotype).

*Figure 5 continued on next page*

*Figure 5 continued*

The online version of this article includes the following source data and figure supplement(s) for figure 5:

**Source data 1.** Length and positioning of Nav channels in GABAergic axons of control and 4.1B KO mice.

**Figure supplement 1.** Myelination and positioning of Nav channels are altered in 4.1B-deficient GABAergic axons.

similar decrease in Kv1 clustering at juxtaparanodes in Lhx6-positive axons as compared with control mice (*Figure 6H, I*). In summary, deficiency in each of the Kv1-associated proteins, Contactin2, Caspr2, and 4.1B induced a pronounced decrease in the juxtaparanodal Kv1 clustering along myelinated GABAergic axons in the hippocampus.

## The excitability of Lhx6-positive interneurons is differently affected in the stratum oriens and stratum pyramidale of 4.1B KO mice

We showed that the 4.1B KO mice exhibited a selective loss of myelin coverage of the inhibitory axons crossing the stratum radiatum. The PV axons crossing the stratum radiatum that may belong to bistratified interneurons were dysmyelinated whereas PV axons of basket cells seemed to be more preserved in the stratum pyramidale. These observations led us to further question whether the phenotypes observed could be associated with physiological changes in hippocampal interneurons. We first recorded the activity of Lhx6Cre;tdTomato-positive cells in the stratum pyramidale of control and 4.1B KO mice (*Figure 7A–G*; *Figure 7—source data 1*). Different subtypes of Lhx6-interneurons are located in the stratum pyramidale as schematized in *Figure 7A*, including the fast-spiking basket cells innervating the soma of pyramidal cells, axo-axonic cells contacting their AIS, and bistratified cells contacting their basal and apical dendrites (*Tricoire et al., 2011*). We performed whole-cell patch-clamp in current-clamp configuration recordings on P30-P55 acute slices and selected fast-spiking interneurons displaying a typical continuous non-stuttering discharge (*Hu et al., 2014*) with a frequency above 50 Hz under injection of a 200 pA current intensity (*Figure 7B*). As shown by the *F-I* curve during injection of current steps of 100 pA increments, the firing frequencies were unchanged in 4.1B KO compared to control mice (p=0.4128, two-way ANOVA; *Figure 7B and C*). The resting membrane potential (Vrest) (–60.50±1.99 mV in 4.1B KO, vs –57.67±3.26 mV in control, p=0.6748 using Mann-Whitney test) and the input resistance (Rinput) (346.20±37.77 MΩ in 4.1B KO vs 360.11±52.46 MΩ in controls, p=0.8421 using Mann-Whitney test) were not changed (*Figure 7D and E*). Similarly, the rheobase (28.80±6.33 pA in 4.1B KO vs 36.67±5.00 pA in controls, p=0.1305, Mann-Whitney test; *Figure 7F*) and afterhyperpolarization (AHP) area (–276.69±42.71 mV.ms in 4.1B KO vs –185.39±54.05 mV.ms in controls, p=0.2775, Mann-Whitney test) were not significantly affected (*Figure 7G*). Overall, we found no significant modification in the intrinsic electrophysiological properties of fast-spiking PV cells (*Table 1*; *Figure 7—source data 1*). Thus, the fast-spiking PV cells, which axons only displayed mild reduction of myelin in the stratum pyramidale, do not exhibit change in their excitability.

To investigate the electrophysiological properties of SST cells, we next recorded Lhx6Cre;tdTomato-positive interneurons in the stratum oriens. As shown in the schematic illustration, different subtypes of Lhx6-interneurons are located in the stratum oriens including SST O-LM cells innervating distal dendrites and bistratified cells connecting basal and apical dendrites of pyramidal cells (*Figure 7H*). In addition, SST GABAergic projection cells have long-running axons in the stratum radiatum (*Gulyás et al., 2003*; *Jinno et al., 2007*) that may be also dysmyelinated. The O-LM cells are the most abundant subtype of Lhx6-interneurons in the stratum oriens and display continuous discharge activity in contrast to the bistratified cells (*Tricoire et al., 2011*). Our immunostaining results indicated that the SST axons crossing straightly the stratum radiatum, likely O-LM axons, presented reduced myelin coverage. Here, we selected Lhx6-positive cells that displayed a non-stuttering and continuous pattern of discharge during spike trains (500ms) as illustrated by traces for control and 4.1B KO mice upon 120 pA current injection (*Figure 7I*). We observed a marked decrease in the mean discharge frequency of 4.1B-deficient cells as indicated by the *F-I* curve depending on injected current steps of 10 pA increments (p<0.0001, two-way ANOVA). Notably, the discharge frequency was significantly decreased upon 100, 110, 120, and 130 pA stimulation (p=0.0375, p=0.0176, p=0.0022, p=0.014, respectively; two-way ANOVA and Sidack post-test; *Figure 7J*; *Figure 7—source data 2*). The resting membrane potential (Vrest) was unchanged (–57.59±1.65 mV in 4.1B KO vs –62. 17±2.93 mV

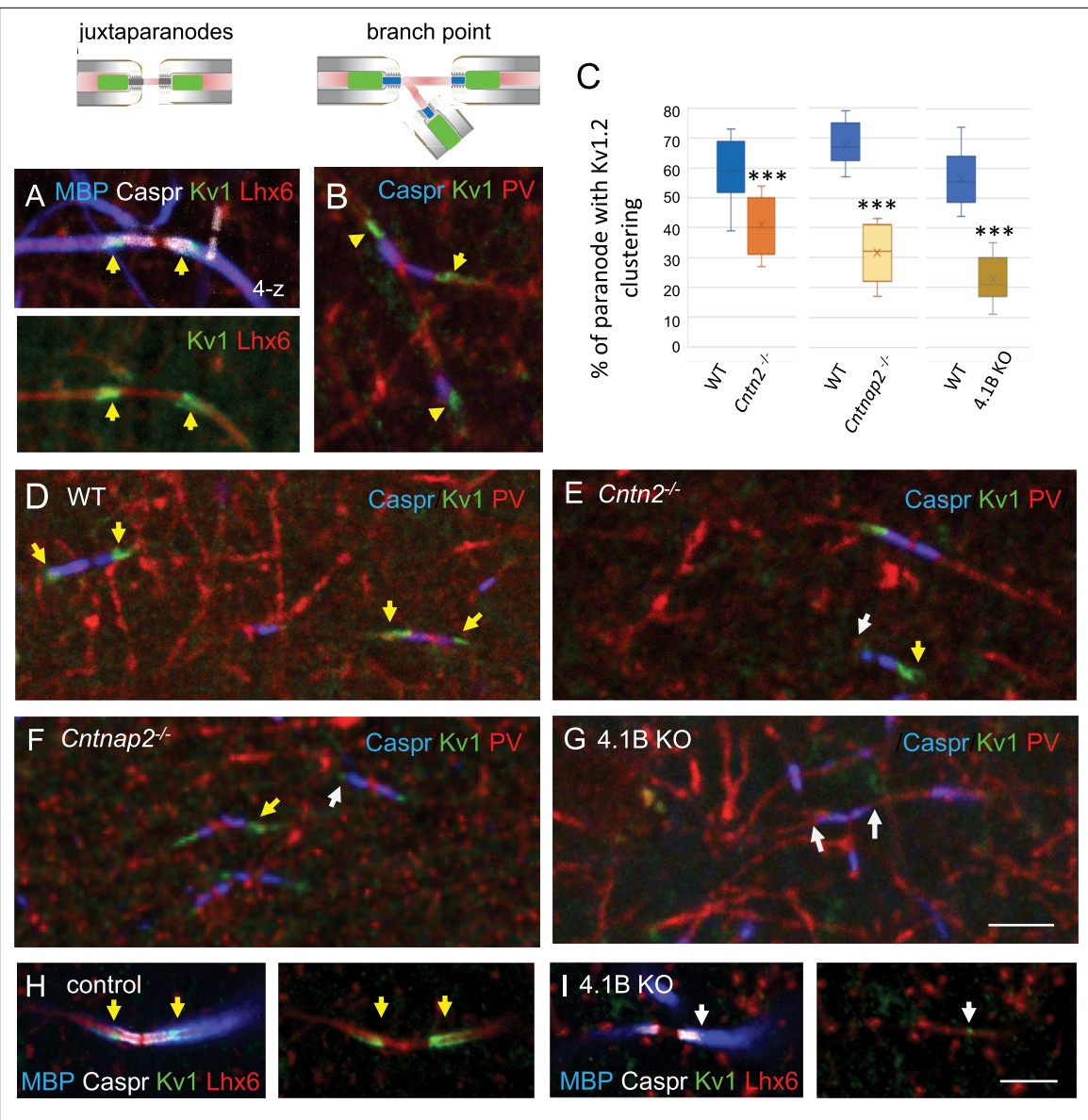

**Figure 6.** The clustering of juxtaparanodal Kv1 channels is reduced in GABAergic axons from *Cntn2*<sup>-/-</sup>, *Cntnap2*<sup>-/-</sup>, and 4.1 KO mice. Immunostaining for Kv1.2 channels in GABAergic axons of the stratum radiatum in CA1 at P70. (**A**) Juxtaparanodal clustering of Kv1 channels in *Lhx6-Cre;tdTomato* control mouse immunostained for MBP (blue), Caspr (white) and Kv1.2 (green). tdTomato expressed in Lhx6-positive cells is in red. (**B**) Juxtaparanodal clustering of Kv1 channels (green) in wild-type at PV axon branch point (red) with paranodes stained for Caspr (blue). (**C**) Quantification of the percentage of paranodes associated with Kv1.2 clustering in the PV axons of the CA1 stratum radiatum. Mutant mice are compared with their respective controls. Means ± SEM of 7–11 ROIs from 3 mice/genotype. Significant difference by comparison with wild-type: *** p<0.001 using the Mann-Whitney test. (**D–G**) Immunostaining for Kv1.2 (green) in wild-type (**D**), *Cntn2*<sup>-/-</sup> (**E**), *Cntnap2*<sup>-/-</sup> (**F**), and 4.1B KO (**G**) PV axons (red) with paranodes stained for Caspr (blue). Yellow arrows point to juxtaparanodal clustering of Kv1 in control or mutant mice and white arrows indicate the lack of proper Kv1 clustering in mutants. Loss of Kv1 clustering is also observed in 4.1B KO crossed with *Lhx6-Cre;tdTomato* mice (I compared to control in H). Bar: 5 µm.

The online version of this article includes the following source data for figure 6:

**Source data 1.** Juxtaparanodal clustering of Kv1 channels in PV axons of wild-type, *Cntn2*<sup>-/-</sup>, *Cntnap2*<sup>-/-</sup>, and 4.1 KO mice.

in controls, p=0.2527, Mann-Whitney test; *Figure 7K*). Hyperpolarizing current steps were injected for analyzing input resistance (Rinput) that was significantly decreased by 22% (278.94±21.12 MΩ in 4.1B KO vs 357.33±30.30 MΩ in controls, p=0.0236, Mann-Whitney test; *Figure 7L*). The rheobase was not significantly changed (52.65±5.79 pA in 4.1B KO vs 38.25±5.73 pA in controls, p=0.1221, Mann-Whitney test; *Figure 7M*). The intrinsic parameters of APs were also analyzed showing an increased AHP area (−526.59±53.50 mV.ms in 4.1B KO vs −351.29±58.99 mV.ms in controls, p=0.0380,

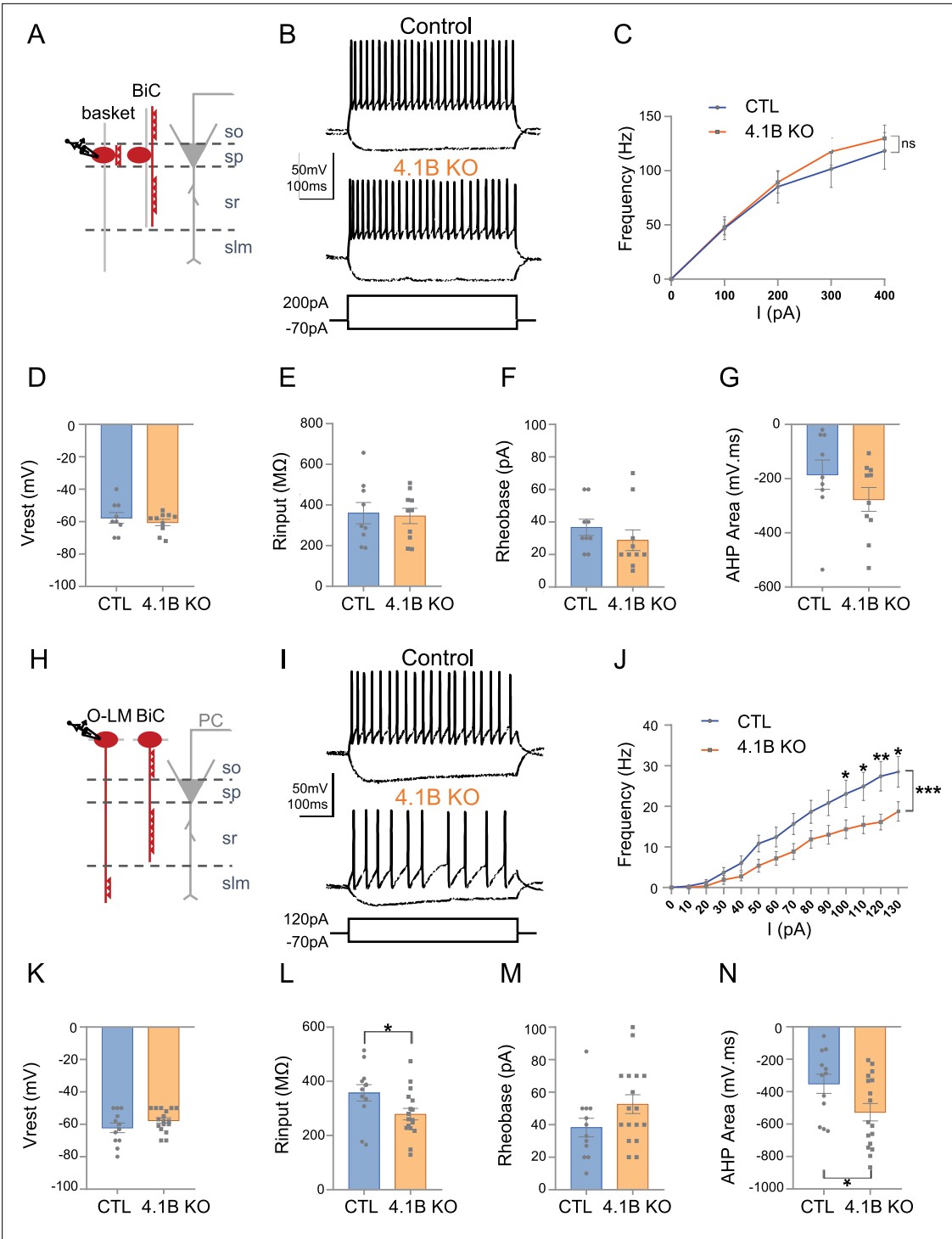

**Figure 7.** The excitability of SST interneurons in the stratum oriens is selectively decreased in 4.1B KO mice. (**A–G**) Patch-clamp recordings of Lhx6-positive PV fast-spiking interneurons in the stratum pyramidale (SP). (**A**) Schematic illustration of basket and bistratified (BiC) cells in CA1 located in the SP and innervating the soma and dendrites of a pyramidal cell (PC). (**B**) Typical fast-spiking activity evoked by a 200 pA stimulation during 500ms showing similar firing patterns of 4.1B-deficient and control PV cells. (**C**) F-I relationship of mean spike frequency depending on current intensities is not different between the genotypes. The resting membrane potential (Vrest) (**D**), input resistance (Rinput) (**E**), rheobase (**F**) and afterhyperpolarization (AHP) area (**G**) are not affected in 4.1B KO mice. Means ± SEM. Detailed statistical analysis using Mann-Whitney test, see *Table 1*. (**H–N**) Patch-clamp recordings of Lhx6-positive interneurons in the stratum oriens (SO) from control and 4.1B KO mice. (**H**) O-LM and bistratified cells in CA1 located in the SO and innervating PC dendrites. (**I**) Example of continuous spike train discharge elicited by a 120 pA depolarizing current pulse during 500ms and voltage response to a –70 pA hyperpolarizing current in control and 4.1B KO mice. (**J**) F-I curves. The mean frequency is significantly reduced in

*Figure 7 continued on next page*

*Figure 7 continued*

4.1B-deficient inhibitory neurons compared to controls (two-way ANOVA, ***p<0.0001, **p<0.01, *p<0.05). (**K**) The resting membrane potential is not affected. (**L**) The Rinput is significantly decreased and the rheobase (**M**) is not changed. (**N**) The AHP area of individual AP is significantly increased (*p<0.05, Mann-Whitney test). For a detailed statistical summary of intrinsic parameters, see *Table 1* and *Table 2* (n=6–9 mice/genotype).

The online version of this article includes the following source data for figure 7:

**Source data 1.** Electrophysiological properties of Lhx6-positive PV fast-spiking interneurons in the stratum pyramidale from control and 4.1B KO mice.

**Source data 2.** Electrophysiological properties of Lhx6-positive interneurons in the stratum oriens from control and 4.1B KO mice.

Mann-Whitney test; *Figure 7I and N* and *Table 2*; *Figure 7—source data 2*). These data indicated that the dysmyelinating phenotype of 4.1B KO mice in SST cells was associated with a selective decrease in their excitability.

## Structural modification of the Axon Initial Segment of SST interneurons in 4.1B KO mice

The efficacy of spike generation is related to the structural organization of AIS, which could be modified in dysmyelinated axons. We investigated whether the decreased excitability in SST cells may be due to differences in the localization of AP initiation site. We first used the method of multiple derivatives of the somatic AP signal that allows to visualize a separation of the axonal and somatodendritic APs (*Figure 8A and B*). Our results showed that 58% of control stratum oriens interneurons (n=12, 9 mice) had a stationary inflection in the rising phase of the AP (*Figure 8A2 and C* top pie chart). We calculated the second ($d^2V/dt^2$) and third ($d^3V/dt^3$) derivatives to better illustrate separation of two inflection points (*Figure 8A3*, arrow in A4) as an indication of the distant site of the AP initiation from the soma. In contrast, 65% of 4.1B-deficient interneurons (n=17, 8 mice) did not display stationary inflection (*Figure 8B2–B and C* bottom pie chart). This statistically significant difference (p=0.0017, Fisher's exact test) suggests that the AP was initiated closer to the soma in 4.1B-deficient neurons than in control ones. This could be due to a modification of AIS geometry such as a shortening or a displacement relative to the soma.

We thus evaluated whether the loss of myelin in hippocampal SST interneurons could be associated with remodeling of the AIS using immunostaining for AnkyrinG. We observed that the AIS of SST interneurons in the stratum oriens was significantly shorter (*Figure 8D–F*; *Figure 8—source data 1*) in 4.1B KO by comparison with wild-type mice (21.93±0.46 µm in 4.1B KO vs 25.26±0.57 µm in wild-type; p<0.0001, Mann-Whitney test, n=30 interneurons, 3 mice/genotype). The AIS onset measured

**Table 1.** Intrinsic electrophysiological properties of fast-spiking Lhx6-interneurons in the stratum pyramidale.

| | Control (mean ± SEM, n=9) | 4.1B KO (mean ± SEM, n=10) | MW test, p value |
|---|---|---|---|
| **Passive properties** | | | |
| Resting membrane potential (mV) | 57.67±3.26 | 60.50±1.99 | MW, *0.6748* |
| Input resistance (MΩ) | 360.11±52.46 | 346.20±37.77 | MW, *0.8421* |
| Rheobase (pA) | 36.67±5.00 | 28.80±6.33 | MW, *0.1305* |
| **Action potential parameters** | | | |
| Amplitude (mV) | 60.42±3.47 | 56.54±2.79 | MW, *0.3882* |
| Area (mV.s) | 58.69±6.00 | 56.44±4.74 | MW, *0.4967* |
| Half-width (ms) | 0.92±0.07 | 0.94±0.06 | MW, *0.9838* |
| Threshold (mV) | 46.79±2.28 | 46.60±1.78 | MW, *0.9682* |
| AHP area (mV.ms) | 185.39±54.05 | 276.69±42.71 | MW, *0.2775* |
| AHP peak (mV) | 10.35±1.85 | 10.65±1.10 | MW, *0.6607* |

Statistical differences between groups were calculated with a Mann-Whitney test (MW). Control: n=9 mice; 4.1B KO: n=6 mice.

**Table 2.** Intrinsic electrophysiological properties of Lhx6-interneurons in the stratum oriens.

|  | Control (mean ± SEM, n=9) | 4.1B KO (mean ± SEM, n=17) | MW test, p value |
|---|---|---|---|
| Passive properties |  |  |  |
| Resting membrane potential (mV) | 62.17±2.93 | 57.59±1.65 | MW, *0.2527* |
| Input resistance (MΩ) | 357.33±30.30 | 278.94±21.12 | MW, *0.0236*\* |
| Rheobase (pA) | 38.25±5.73 | 52.65±5.79 | MW, *0.1221* |
| Action potential parameters |  |  |  |
| Amplitude (mV) | 65.34±1.84 | 64.06±1.94 | MW, *0.6788* |
| Area (mV.s) | 111.44±17.87 | 104.80±8.93 | MW, *0.8788* |
| Half-width (ms) | 1.61±0.25 | 1.63±0.16 | MW, *0.5929* |
| Threshold (mV) | 45.69±1.83 | 43.65±1.43 | MW, *0.5558* |
| AHP area (mV.ms) | 351.29±58.99 | 526.59±53.50 | MW, *0.0380*\* |
| AHP peak (mV) | 12.57±1.79 | 16.24±1.25 | MW, *0.1280* |

Statistical differences between groups were calculated with a Mann-Whitney test (MW). Control: n=9 mice; 4.1B KO: n=8 mice.

from the soma or the primary dendrite was not modified (3.8±0.27 µm in 4.1B KO vs 4.13±0.40 µm in wild-type; p=0.6509, Mann-Whitney test; *Figure 8G*; *Figure 8—source data 1*). Taken together, our results indicated that dysmyelination in 4.1B KO mice caused a shortening of AIS and a net decrease in the intrinsic excitability of SST cells.

## Inhibitory inputs onto CA1 pyramidal neurons are influenced by dysmyelinated 4.1B KO phenotype

We then asked the question of the role of interneuron myelination in the proper inhibitory synaptic transmission on pyramidal cells. First, we recorded spontaneous inhibitory postsynaptic currents (sIPSCs) on pyramidal cells in APV (40 µM) and CNQX (10 µM) condition (*Figure 9A and B*). We showed a significant increase in sIPSCs peak amplitude in the 4.1B KO mice (45.61±2.93 pA in 4.1B KO vs 31.51±0.88 pA in controls; p=0.0022, Mann-Whitney test, n=6 cells from 4 to 5 mice). The frequency of sIPSCs also tended to be increased (4.97±1.02 Hz in 4.1B KO vs 2.76±1.13 Hz in controls, p=0.1320, Mann-Whitney test; *Figure 9C*; *Figure 9—source data 1*). Interestingly, the occurrence probability of small amplitude sIPSCs was decreased in 4.1B KO mice (p<0.0001, Kolmogorov-Smirnov test; *Figure 9D*; *Figure 9—source data 1*). To support this observation, we next examined the rise time and amplitude of 20 isolated sIPSCs per cell. The cumulative distribution showed that the occurrence probability of sIPSCs with slow rise time was decreased in 4.1B KO by comparison with control mice (p<0.0001, Kolmogorov-Smirnov test; *Figure 9E*). In addition, both genotypes showed a significant correlation between the rise time and amplitude of sIPSCs, as shown in *Figure 9F* (p=0.0077 in controls and p=0.0072 in 4.1B KO, Pearson correlation test). Linear regression analysis of the relationship between amplitude and rise time of sIPSCs showed that while the slopes were comparable between genotypes (Null hypothesis test, p=0.9791), there was a significant difference in their elevations towards a faster rise time in 4.1B KO mice (Null hypothesis test, *F*=13.55, DFn = 1, DFd = 237, p=0.0003). Thus, events with small amplitude and slower rise time could be less abundant among the sIPSCs in 4.1B KO mice. To further examine the properties of inhibitory synaptic inputs, we recorded miniature inhibitory postsynaptic currents (mIPSCs) on pyramidal cells as AP-independent inhibition. We found that neither the mean amplitude (28.29±2.33 pA in 4.1B KO mice vs 24.57±2.68 pA in controls; p=0.2224, Mann-Whitney test, n=9 cells from 3 to 5 mice) nor the mean frequency were affected in 4.1B KO mice (2.41±0.47 Hz in 4.1B KO mice vs 2.73±0.51 Hz in controls, p=0.5457, Mann-Whitney test). This data on mIPSCs indicated that the 4.1B mutation may not modify the inhibitory pre-synaptic terminals connecting pyramidal cells.

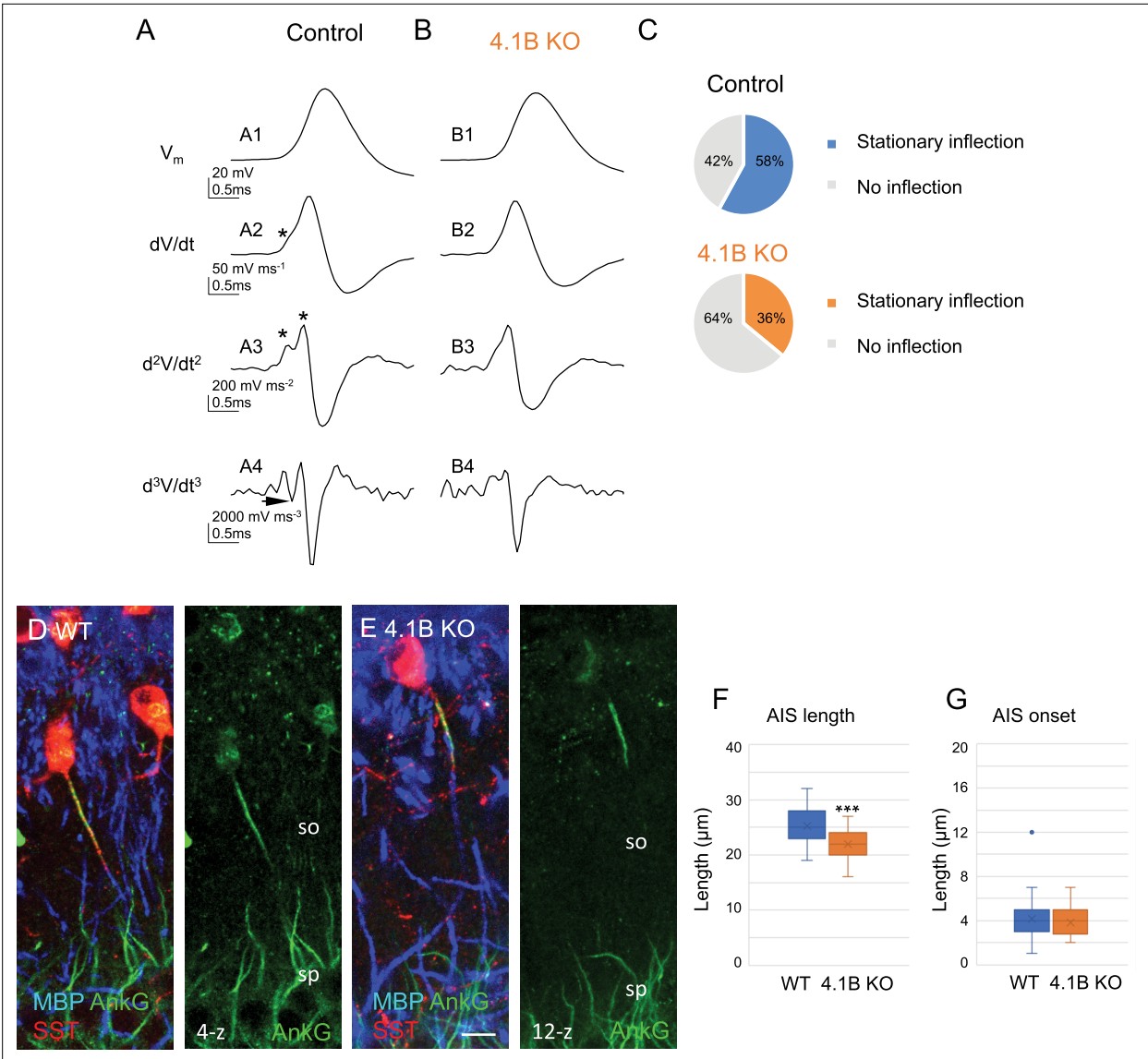

**Figure 8.** Structural modification of the AIS of SST interneurons in 4.1B KO mice. (**A–C**) Analysis of somatic action potentials (APs). Representative examples illustrating the voltage waveform of the AP (A1 and B1; in mV) from a control (**A1**) and a 4.1B KO neurons (**B1**). Below are presented the first (A2 and B2; dV/dt, in mV/ms), second (A3 and B3; d²V/dt², in mV/ms²), and third derivatives (A4 and B4; d³V/dt³, in mV/ms³). The stationary inflection in the d²V/dt² trace is indicated with asterisks. The third derivative allows to detect the second inflection point when the trace reaches or goes under zero (arrow). (**C**) We observed a second inflection point in 58% of control (top pie graph) and 35% of 4.1B-deficient cells (bottom pie graph). This difference in the number of cells displaying an inflection point (asterisks in A2 and A3) is significant (p=0.017, Fisher's exact test) (n=12 cells from 9 control mice and n=17 cells from 8 4.1B KO mice). This could be due to the AIS shortening in 4.1B-deficient SST interneurons. (**D, E**) Hippocampal sections of wild-type (**D**) and 4.1B KO (**E**) mice at P35 immunostained for MBP (blue), SST (red) and AnkyrinG (green). Confocal images with maximum intensity of z-steps of 0.54 µm. (**F**) Length of AIS measured for AnkyrinG immunostaining of SST interneurons in the stratum oriens (SO). Significant difference by comparison with wild-type: *** p<0.0001 using Mann-Whitney test. (**G**) Distance of AIS onset from the soma is not changed between wild-type and 4.1B KO mice (p=0.6509, Mann-Whitney test; n=30 cells in wild-type and 4.1B KO mice; 3mice/genotype). Bar: 10 µm in (**D, E**).

The online version of this article includes the following source data for figure 8:

**Source data 1.** Length and onset of AIS measured for AnkyrinG immunostaining of SST interneurons in the stratum oriens of wild-type and 4.1B KO mice.

The increased amplitude of sIPSCs in 4.1B KO mice could reflect an enhanced proximal and a reduced distal GABAergic inhibition onto pyramidal cells. Taken together, these results suggest that the distal inhibitory inputs onto pyramidal cells may be reduced as a consequence of the dysmyelination and decreased excitability of SST cells.

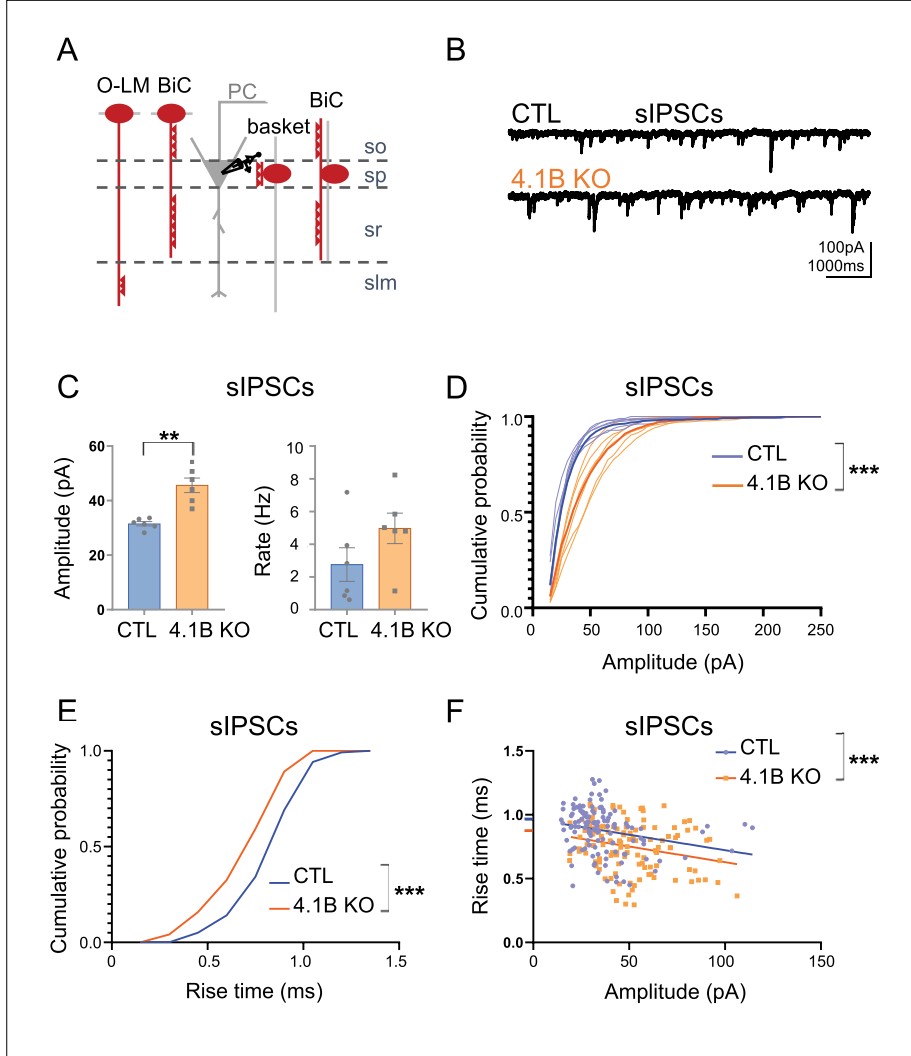

**Figure 9.** Inhibitory inputs onto CA1 pyramidal neurons are affected in 4.1B KO mice. (**A**) Schematic illustration of inhibitory inputs onto the soma, proximal and distal dendrites of pyramidal cells. (**B**) Example traces of sIPSCs onto CA1 pyramidal neurons of control and 4.1B KO mice. (**C**) The amplitude of sIPSCs is significantly increased in 4.1B KO mice (**p=0.0022) and the frequency is not affected (p=0.1320); 15–400 pA events analyzed during 3 min. Statistical values were performed with a Mann-Whitney test. (**D**) Cumulative probability for the amplitude of sIPSCs (3 min, 6 cells and 4–5 mice/genotype) showing a significant different distribution between genotypes (Kolmogorov-Smirnov test, ***p<0.0001). Note that the probability of small amplitude events is reduced in 4.1B KO mice. (**E**) Cumulative probability for the rise time of sIPSCs (20 isolated events/cell, 6 cells and 4–5 mice/genotype) showing a significant difference in the distribution between genotypes towards faster rise time in 4.1B KO mice (Kolmogorov-Smirnov test, ***p<0.0001). (**F**) Relationship between rise time and amplitude of events in E and linear regression for each genotype; this shows a significant difference between the elevations of the linear regressions towards faster rise time in 4.1B KO mice (Null hypothesis test, ***p=0.0003).

The online version of this article includes the following source data for figure 9:

**Source data 1.** Amplitude and frequency of sIPSCs onto CA1 pyramidal neurons in the control and 4.1B KO mice.

## Discussion

Our results describe a genetic model of selective myelin loss in the hippocampus. We show that the extent of myelination of PV and SST axons is severely reduced in the stratum radiatum. Deficiency in the scaffolding protein 4.1B also impairs Kv1 clustering at juxtaparanodes and proper localization of Nav at the nodes of Ranvier. The excitability of the SST cells in the stratum oriens is decreased while

the properties of the fast-spiking PV interneurons in the stratum pyramidale are unchanged. Our data suggest that this excitability defect reduces the distal inhibitory drive onto pyramidal cells.

## 4.1B is implicated in the myelin coverage of hippocampal inhibitory axons

Despite the well-established physiological importance of myelination to speed axonal conduction, relatively little is known about its significance for GABAergic local interneurons. Recent reports have highlighted the fact that the different subtypes of GABAergic neurons could be myelinated and suggested that oligodendrocytes can recognize the class identity of individual types of interneurons that they target (*Zonouzi et al., 2019*). PV- and to a lesser extent SST-expressing cells in the CA1 mouse hippocampus are frequently myelinated on their proximal axonal segments, independently of their subtype identity (*Micheva et al., 2016*; *Stedehouder et al., 2017*). The sparse myelination of inhibitory ramified axons implies the presence of heminodes along mature axons. One inherent feature that limits myelin sheath growth and determines node position is the branching pattern. An intriguing question is whether axonal cues may be instructive to determine the extent of myelin sheath coverage. It is known that hippocampal inhibitory axons can form pre-nodal clusters of Nav channels along premyelinated axons (*Bonetto et al., 2019*; *Freeman et al., 2015*). At a later stage of development, such prenodes of Nav/Neurofascin/AnkyrinG may indicate positions for limiting myelin sheath extension (*Malavasi et al., 2021*; *Thetiot et al., 2020*; *Vagionitis et al., 2022*). During development, components of the nodes of Ranvier accumulate adjacent to the ends of myelin sheaths to form heminodes prior to forming nodes. The developing myelin sheaths may stop growing when they meet a prenodal cluster, which would be instructive for the position of the node. Indeed, mutation of the nodal CAM Neurofascin-186 has been reported to increase internodal distance (*Vagionitis et al., 2022*). Here, we observed that deletion of 4.1B reduces the mean internodal length indicating that myelin sheath elongation is restricted in the mutant mice. 4.1B-deficiency strongly reduces myelin ensheathment of both PV and SST axons in the stratum radiatum, by 58 and 63%, respectively. The scaffolding 4.1B, which mediates the linkage between Caspr and the ßII-spectrin may be implicated as a paranodal cue modulating the myelin coverage of GABAergic axons. The association of 4.1 proteins with their transmembrane receptors and the actin-spectrin cytoskeleton can be negatively modulated by phosphorylation (*Wang et al., 2014*) so that 4.1B may act as a clutch to promote myelin sheath elongation depending on axonal signaling. Whether the paranodal cytoskeleton including 4.1B may be implicated in adaptive myelination deserves further investigation.

The localized effect of 4.1B-deficiency on the myelination of hippocampal interneurons is intriguing. The stratified organization of the hippocampus likely reveals the selective role of 4.1B in promoting myelin sheath elongation in inhibitory interneurons. In the stratum radiatum, we evaluated that as much as 75% of the myelinated axons are Lhx6-positive. This is in accordance with other studies indicating that the fraction of myelin ensheathing GABAergic neurons is reaching up 80% in the stratum pyramidale and stratum radiatum (*Stedehouder et al., 2017*). We did not detect any massive loss of myelin immunostaining in other brain areas of 4.1B KO mice as illustrated in the somato-sensory cortex (*Figure 1—figure supplement 2*). However, a precise quantitative study would be required to examine whether selective dysmyelination of GABAergic interneurons is induced by 4.1B-deficiency throughout cortical layers and in particular, in layer2/3, in which nearly half of the myelin ensheaths inhibitory neurons (*Micheva et al., 2016*; *Stedehouder et al., 2017*). On the other hand, the band 4.1 transcripts undergo extensive alternative splicing and belongs to a family of proteins including 4.1 G, 4.1 N, and 4.1 R proteins expressed in brain (*Buttermore et al., 2011*; *Ivanovic et al., 2012*; *Kang et al., 2009*). Therefore, we can hypothesize that the cell-type specific effect of 4.1B deletion may rely on compensatory mechanisms such as upregulation of 4.1 G or 4.1 R, as shown in the PNS (*Einheber et al., 2013*; *Horresh et al., 2010*).

## 4.1B is required for the assembly of nodal and juxtaparanodal domains in myelinated inhibitory axons

In addition to the severe reduction of myelin sheath coverage, 4.1B-deficiency induces alteration of juxtaparanodal and nodal ion channels. The scaffolding protein 4.1B is linked to the Caspr2/Contactin2 cell adhesion complex at juxtaparanodes (*Brivio et al., 2017*; *Buttermore et al., 2011*; *Zhang et al., 2013*; *Zonta et al., 2008*). As previously reported in the PNS and CNS (*Buttermore*

*et al., 2011; Cifuentes-Diaz et al., 2011; Einheber et al., 2013; Horresh et al., 2010*), juxtaparanodal Kv1 clustering is strongly reduced in 4.1B-deficient GABAergic axons of the hippocampus. We observed a similar alteration in Contactin2- and Caspr2-deficient GABAergic axons, although some residual expression of Kv1.2 can be detected at the juxtaparanodes of the mutant mice. The Kv1 channels are localized under the compact myelin and their function is still elusive under physiological conditions (*Pinatel and Faivre-Sarrailh, 2020*). On the other hand, protein 4.1B binds Caspr at paranodes where it organizes the boundary with the nodal Nav channels. Strikingly, the nodal and heminodal Nav channels are no longer juxtaposed to the paranodes in the adult 4.1B KO mice, as previously reported in the developing spinal cord and likely as a consequence of the disorganization of the spectrin cytoskeleton (*Brivio et al., 2017*).

Whether the loss of myelin in the 4.1B mutant influences ion channel expression at the AIS is another intriguing question. We observed structural modification of the AIS of SST cells in the stratum oriens. The length of the AIS as measured with AnkyrinG is significantly shorter in 4.1B KO by comparison with wild-type mice, whereas its onset from the soma or the primary dendrite is not modified. Protein 4.1B is not required for the stabilization of AnkyrinG at the AIS as shown in cultured hippocampal interneurons from 4.1B KO mice (*Bonetto et al., 2019*). However, 4.1B interacting with NuMA1 inhibits endocytosis of AIS membrane proteins and is required for AIS assembly, but not maintenance, as shown by shRNA silencing 4.1B expression at early stage of hippocampal culture (*Torii et al., 2020*). Therefore, we propose that the AIS shortening in 4.1B-deficient SST interneurons may not be a primary event but rather a result of an adaptation to the reduction of myelin axonal coverage. Supporting this notion, previous studies have reported an adaptive response of the AIS of cortical pyramidal neurons in the cuprizone chemical model of demyelination. Specifically, it was observed that the length of the AIS is reduced together with a more proximal site of the onset. These changes reduce the AIS excitability suggesting a compensatory mechanism to ectopic action potentials generated in demyelinated axons (*Hamada and Kole, 2015*).

## Class-specific effect of dysmyelination on the excitability of hippocampal interneurons

A defining feature of inhibitory interneuron subsets is their precise axonal arborisation whereby inhibitory synapses target specific subdomains of pyramidal neurons. PV basket cells innervate the perisomatic region and the SST O-LM interneurons connect the distal dendrites of pyramidal neurons. We observed a severe reduction of myelin in the stratum radiatum, whereas myelination appeared slightly affected in the stratum pyramidale. Such a layer-specific dysmyelination may indicate that the O-LM and bistratified interneurons may be more affected than the basket cells. The myelinated pattern of O-LM SST interneurons is highly visible since they extend myelinated axons from the stratum oriens straight across the stratum radiatum. The sheath coverage of O-LM axons is reduced by 55% in the 4.1B KO by comparison with wild-type mice.

We asked whether dysmyelination may induce changes in the neuronal excitability of interneuron subtypes innervating the different layers of the CA1 hippocampus. In this context, we showed that PV cells in the stratum pyramidale displayed a similar pattern of fast-spiking discharge in 4.1B KO than in control mice. Even if the rheobase tended to decrease suggesting that 4.1B-deficient PV cells could be more excitable than controls, we found no change in the other intrinsic parameters measured at the soma. In contrast, the dysmyelinated phenotype of stratum oriens SST interneurons is associated with lower excitability in 4.1B KO mice. This was correlated with a significant decrease in the input resistance and a significant increase in the AHP area. Such electrophysiological modifications may reflect changes in ion channel expression. Nevertheless, we cannot exclude that the reduction in excitability of 4.1B-deficient SST neurons could also result from the change in their morphometry such as cell body size. Interestingly, in cortical pyramidal neurons, demyelination induced by cuprizone causes the restructuring of AIS and reduces excitability at this site. Acute demyelination leads to a more proximal onset of AIS without a change in the length of ßIV spectrin expression. However, the AIS of these acutely demyelinated axons display a reduced length of Nav1.6 channel expression and extended Kv7.3 channel expression at the distal site (*Hamada and Kole, 2015*). The $I_M$ current comprised of $K_v7$ channels is known to control the interevent intervals and is particularly effective in influencing firing frequency of O-LM cells (*Lawrence et al., 2006*). Therefore, one can hypothesize that the increase of AHP area in 4.1B-deficient SST cells could be due to a change in Kv7 expression at

the AIS. In addition, we showed that the length of the AIS is significantly reduced without any change in its distance from the soma, which could lead to a decrease in excitability. Indeed, structural characteristics of the AIS, such as the length and/or distance to the soma, strongly affect the excitability and firing behavior of neurons (*Kole and Stuart, 2012*; *Kuba, 2012*; *Yamada and Kuba, 2016*). Along this line, the derivative analysis suggests that the AP is initiated closer to the soma in 4.1B-deficient SST neurons than in controls. In addition, protein 4.1B is associated in a complex with the Kv1 channels along the axon. The clustering of Kv1 channels is strongly reduced at juxtaparanodes of 4.1B-deficient mice and as a consequence, Kv1 can be diffuse along the axon to participate in the modulation of intrinsic excitability (*Rama et al., 2017*).

Functionally, it is well established that PV interneurons are involved in proximal inhibition while SST and also some CCK interneurons take part in distal dendritic inhibition (*Somogyi and Klausberger, 2005*; *Udakis et al., 2020*). Thus, we investigated whether the inhibitory inputs could be modified onto the pyramidal cells. We showed that the occurrence probability of small amplitude and slower rise time events among sIPSCs are reduced in 4.1B KO mice. This suggests that either the distal inhibitory inputs onto pyramidal cells may be decreased as a consequence of the reduced excitability of SST cells or that the proximal inhibition by PV cells may be increased. Nevertheless, this last hypothesis is unlikely since we did not observe an increase in the excitability of fast-spiking PV interneurons. The mIPSCs recorded in pyramidal cells did not differ in amplitude and frequency between control and 4.1B-deficient neurons suggesting that GABAergic receptors at the post-synaptic site and the number of presynaptic terminals are similar between both genotypes. This last result indicates that dysmyelination of inhibitory axons in CA1 may not be associated with structural synaptic defects, but with a reduction of distal inhibitory inputs. This together with the results from immunolabelling suggest that O-LM cells are the main dysmyelinated specific class of interneurons whose excitability may affect the inhibitory neuronal activity.

An important point to keep in view is that myelin sheath, apart from its insulator role, provides metabolic and trophic support in the central nervous system (*Kann et al., 2014*; *Krasnow and Attwell, 2016*). As a possible consequence of impaired trophic support, it has been reported that in cortical neurons from lysophosphatidylcholine demyelinated mice, there is a decrease in mIPSC amplitude and frequency consistent with a selective loss of PV inhibitory synapses (*Zoupi et al., 2021*). Similarly, demyelination of cortical PV basket cells from cuprizone treated mice strongly reduces the density of inhibitory presynaptic terminals onto the soma of pyramidal cells (*Dubey et al., 2022*). In contrast, we did not evidence synaptic alterations associated with dysmyelinated interneurons in 4.1B KO mice. As for long projecting neurons, it is also hypothesized that myelin optimizes the propagation of APs along inhibitory axons. Myelination defects of cortical PV fast-spiking cells are associated with reduced firing frequency and delayed predicted conduction velocity of their APs (*Benamer et al., 2020a*). Indeed, a positive correlation between the degree of axonal myelination and conduction velocity has been reported for cortical PV interneurons (*Micheva et al., 2021*). It has been also shown that the topography of interneuron myelination depends on the axon diameter and interbranch distance (*Stedehouder et al., 2019*). Here, we observed that the unramified axonal segment of O-LM cells crossing the stratum radiatum is ensheathed by myelin throughout. The strong reduction of myelin coverage along O-LM cell axons in 4.1B KO mice should decrease AP conduction velocity. In acute demyelinated cortical axons, conduction velocity of APs is reduced and ectopic APs appear; this is associated with a redistribution of Na$^+$ channels at branch points with increased nodal expression length of Nav1.6 (*Hamada and Kole, 2015*). Here, we show mislocalization and increased length of nodal Nav channels that could induce modifications in membrane capacitance and axial resistance to current flow from the node into the internode (*Arancibia-Cárcamo et al., 2017*). The misdistribution of ion channels implies that AP propagation should be defective along dysmyelinated SST axons in 4.1B KO mice.

Myelination of interneurons may play a critical role in brain functions such as learning and memory (*Bonetto et al., 2021*; *Yang et al., 2020*). A dysfunction of hippocampal inhibitory neurons can alter theta and gamma oscillations involved in navigation performance (*Kalemaki et al., 2018*; *Kunz et al., 2019*; *White et al., 2012*). The hippocampal SST cells have been reported to contribute to learning-induced persistent plasticity and spatial memory consolidation (*Artinian et al., 2019*; *Honoré et al., 2021*). The O-LM cells are modulating the afferents onto pyramidal cell dendrites, facilitating through disinhibition the Schaffer collateral pathway from CA3 and down-regulating the temporo-ammonic

pathway from the entorhinal cortex (*Leão et al., 2012*). In the 4.1B KO mice, the severe loss of myelin in hippocampal interneurons may be associated with a dysfunction in the inhibitory network. A slight apparent disorganization of myelin was also observed in the molecular layer of the dentate gyrus, likely corresponding to inputs from entorhinal cortex. Although we do not exclude a dysmyelination effect of 4.1B-deficiency of extra-hippocampal afferences connecting the hippocampus, our results highlight the role of myelin in SST interneurons in fine-tuning the hippocampal inhibitory drive that might be involved in spatial exploration and memory.

This study may be relevant for understanding the contribution of interneuron demyelination as described in multiple sclerosis. Demyelinated lesions have been reported in the hippocampus of multiple sclerosis patients correlated with cognitive deficits (*Geurts et al., 2007*). GABAergic inter-neurons are selectively vulnerable to demyelination in multiple sclerosis (*Zoupi et al., 2021*). Inhibitory network modifications could lead to an increased incidence of epileptic seizures in multiple sclerosis patients (*Kelley and Rodriguez, 2009*) and are associated with diminished cognitive function (*Nicholas et al., 2016*). Here, we show a subtype-specific role of myelin in hippocampal GABAergic interneurons. This raises the intriguing question of whether subtypes of interneurons may be specifically affected in demyelinating diseases.

## Methods
### Animals
The care and use of mice in all experiments were carried out according to the European and Institutional guidelines for the care and use of laboratory animals and approved by the local authority (laboratory's agreement number D13-055-8, Préfecture des Bouches du Rhône). The following mouse strains were used in this study: C57BL/6 mice (Janvier Breeding Centre), previously described *Cntn2$^{-/-}$* mice (*Traka et al., 2003*), *Eph41l3* KO also named 4.1B KO mice (*Cifuentes-Diaz et al., 2011*) and *Cntnap2$^{-/-}$* mice (*Poliak et al., 2003*). Both *Cntn2* KO and *Eph41l3* KO mice were crossed with *Lhx6-Cre* (strain 026555) and *Rosa26$^{Ai14}$* (strain 007908) mouse strains (Jackson Laboratory, Bar Harbor, ME, USA) to obtain mice with the following genotypes *Cntn2$^{-/-}$;Lhx6-Cre;Rosa26$^{Ai14}$* and *Eph41l3;Lhx6-Cre;Rosa26$^{Ai14}$* expressing tdTomato. Fixed brains from *Cntnap2* KO obtained from the Jackson laboratory (strain 017482) were a gift from Dr. N. Noraz (NeuroMyogene Institute, Lyon, France). Male and female animals were used indifferently for histological analyses and electrophysiological recordings.

### Immunofluorescence staining, confocal microscopy, and image analysis
Rabbit antiserum against protein 4.1B (*Cifuentes-Diaz et al., 2011*) and rabbit anti-Caspr (*Bonnon et al., 2007*) were described previously. The following commercial primary antibodies were used: rabbit anti-Olig2 antibody (AV32753, RRID:AB_1854792) and mouse anti-panNav mAb (S8809, RRID:AB_477552) purchased from Sigma-Aldrich Chimie, (Saint-Quentin-Fallavier, France), rat anti-MBP mAb (ab7349, RRID:AB_305869) from Abcam (Cambridge, UK), goat anti-PV antibody (PVG-214, RRID:AB_2313848) from Swant (Burgdorf, Switzerland), goat anti-SST mAb (sc7819, RRID:AB_2302603) and mouse anti-SST (sc55565, RRID:AB_831726) from Santa-Cruz Biotechnology (Heidelberg, Germany). Mouse anti-ankyrinG (N106/36, RRID:AB_2877524) and anti-Kv1.2 (K14/16, RRID:AB_10674277) mAbs were obtained from NeuroMab (UC Davis/NIH NeuroMab Facility). AlexaFluor-405, 488,–568 and –647-conjugated secondary antibodies were purchased from Molecular Probes (Life Technologies, Courtaboeuf, France).

P25, P30, P35, P70, and P180 mice were deeply anesthetized with a mix of Zoletyl/Domitor and then transcardially perfused with PBS followed by 2% paraformaldehyde in PBS. Brains were removed and placed in the same fixative overnight. Eighty micron-thick vibratome sections were permeabilized and blocked for 1 hr in PBS containing 5% horse serum and 0.3% Triton X-100. Antigen retrieval in 10 mM citrate buffer, pH6, at 90 °C for 30 min was performed before immunostaining with anti-panNav and anti-AnkyrinG antibodies. Floating sections were incubated for 2 days at 4 °C with combinations of the following primary antibodies: goat anti-PV (1:1000), mouse anti-SST (1:200), goat anti-SST (1:500), rabbit anti-Caspr (1:2000), mouse anti-Kv1.2 (1:400), mouse anti-Kv3.1b (1:100), mouse anti-ankyrinG (1:100), mouse anti-panNav (1:200), rat anti-MBP (1:200), rabbit anti-Olig2 (1:2500), rabbit anti-4.1B (1:2000). Controls for antibody specificity were performed by incubation with secondary antibodies only. Control for the specificity of rabbit anti-4.1B antiserum was performed using brain sections from

4.1B KO mice. Sections were then washed and incubated with the appropriate AlexaFluor-conjugated secondary antibodies (1:500) overnight, washed and mounted on slides with Vectashield mounting medium (Vector Laboratory, Eurobio Scientific, Les Ulis, France). Image acquisition was performed on a Zeiss (Carl-Zeiss, Iena, Germany) laser-scanning microscope LSM780 equipped with 40X1.32 NA oil-immersion objective or 20 X objective. Images of AlexaFluor-stained cells were obtained using the 488 nm band of an Argon laser and the 405 nm, 568 nm and 647 nm bands of a solid-state laser for excitation. Fluorescence images were collected automatically with an average of two-frame scans at airy 1. Maximum intensity projection of images was carried out using ImageJ software (NIH). Images are single confocal sections unless the number of z-steps is indicated.

As shown in *Figure 1I*, the density of paranodes immunolabeled for Caspr was estimated in the different layers (alveus, stratum oriens, stratum pyramidale, stratum radiatum and stratum lacunosum-moleculare) of the CA1 hippocampus at P70. The density of single paranodes (1 doublet of paranodes = 2 single paranodes) was quantified in control and 4.1B-deficient mice using maximum intensity projection images (5-z steps of 2 μm) (4–7 ROI of $10^5$ μm$^2$ from 4 mice/genotype). In *Figure 1J*, the total length of myelinated axons was measured in the stratum oriens (bin1), stratum pyramidale (bin2) and stratum radiatum divided in bins3-7 (40x300 μm) on single confocal sections, n=4 mice/genotype. In *Figure 2K*, we measured the total length of PV-positive myelinated axon in the stratum radiatum at P70 (stacks of 1 μm 5-8z steps; 5 ROIs of $10^5$ μm$^2$ from 3 mice/genotype). In *Figure 2L and M*, the length of myelinated SST axons was measured in the stratum radiatum of control and 4.1B KO mice at P35 (stacks of 1 μm 5-8z steps; 10 ROIs of $10^5$ μm$^2$ from 3 mice/genotype). To analyze the extent of myelin coverage of individual SST-positive axons across the stratum radiatum, only axons with a length >100 μm were taken into account (n=29–35 axons from 3 mice/genotype). Three-D reconstruction of myelinated axons was performed using the Neurolucida Explorer Software (Micro Bright Field, Inc, Williston, VT, USA). In *Figure 4B*, the distribution of the lengths of internodes was evaluated in the CA1 hippocampus of control and 4.1B KO *Lhx6-Cre;tdTomato* mice at P70. Double-immunostaining for MBP and Caspr and confocal imaging were performed (25–50 μm-width z-stacks of 2 μm). Maximum intensity projection images were examined together with single confocal sections using the ImageJ 'synchronize windows' tool. We measured the length of myelin sheaths bordered at each end by a paranode in 4 ROIs of $10^5$ μm$^2$ from 2 mice/genotype (n=275 in control and n=191 in 4.1B KO mice). In *Figure 5*, the length of nodal Nav was measured in the CA1 hippocampus of control *Lhx6-Cre;tdTomato* at P30 and P70. Triple-immunostaining for Caspr, MBP and panNav was performed and Lhx6-positive axons expressing tdTomato were detected with the red channel. The distribution of nodal length was analyzed on confocal images (40 X oil objective, 1 μm 2-5z-steps) from 2 mice/age (P30: n=41, P70: n=64). Next the length of nodal or heminodal Nav channel clusters and the gap between Nav clusters and paranode were measured in the CA1 hippocampus of control and 4.1B KO *Lhx6-Cre;tdTomato* mice at P70 (n>25/genotype from 2 to 3 mice/genotype). In *Figure 6*, the clustering of juxtaparanodal Kv1 channels was estimated in the CA1 hippocampus of wild-type, 4.1B, *Cntnap2*, and *Cntn2* KO mice at P70. The percentage of single paranodes bordered by Kv1.2 immunostaining in PV-positive axons was quantified on 5–10 μm z-stack of 1 μm confocal images. More than 150 single paranodes were analyzed in each genotype in 7–11 ROIs from 3 mice/genotype. Mutant mice were compared with their respective controls (not littermate) in double-blind experiments. In *Figure 8*, the AIS structural parameters (onset, length) were measured in SST neurons of the stratum oriens. Triple-staining for AnkyrinG, SST and MBP was performed at P35 (3 mice/genotype). To encompass AIS onset and distal ending, we selected neurons with SST axonal staining exceeding AnkyrinG AIS staining. The bottom and top sections were carefully determined for z-stack acquisition (20 X objective, 0.54 μm steps). The AIS length was measured on 2D projection taking care that the AIS of SST neurons were orientated in the plane of section (z-width 2–8 μm; means ± SD: 4.74±2.23 in WT; 4.58±1.56 in 4.1B KO). Measurements were performed on stacks using the ImageJ 'synchronize windows' tool.

## Electrophysiological recordings

Mice (P28-55) were deeply anesthetized with Zoletil/Domitor and subsequently decapitated. Brains were quickly removed from the skull and placed into ice-cold oxygenated (95% O2 and 5% CO2) cutting solution containing the following (in mM): 132 choline, 2.5 KCl, 1.25 NaH2PO4, 25 NaHCO3, 7 MgCl2, 0.5 CaCl2, and 8 D-glucose (pH: 7.4, 290–300 mOsm/L). 300 μm-thick coronal brain slices

were prepared using a VT1200S microtome (Leica Microsystems, Wetzlar, Germany). Slices were left at room temperature in oxygenated (95% O2 and 5% CO2) solution of artificial cerebrospinal fluid (aCSF) containing the following (in mM): 126 NaCl, 3.5 KCl, 1.2 NaH2PO4, 26 NaHCO3, 1.3 MgCl2, 2.0 CaCl2, and 10 D-glucose, pH 7.4. Then, slices were transferred into a submerged recording chamber and perfused with oxygenated aCSF at a flow rate of 2–3 ml/min. Recordings were performed under visual guidance using infrared differential interference contrast microscopy (SliceScope Pro 3000 M, Scientifica, Uckfield, UK). tdTomato in neurons was excited by a UV lamp and the fluorescence was visualized using a CCD camera (Hamamatsu, Japan). Whole-cell patch-clamp recordings were made using a Multiclamp 700B amplifier (Molecular Devices, CA, USA), filtered at 2 kHz using the built-in Bessel filter of the amplifier. Data were digitized at 20 kHz with a Digidata 1440 A (Molecular Devices) to a personal computer, and acquired using Clampex 10.1 software (PClamp, Molecular Devices). For current clamp recording mode patch pipettes were pulled from borosilicate glass tubing with resistances of 6–8 MΩ and filled with an internal solution containing the following (in mM): 130 KMeSO$_4$, 5 KCl, 10 4-(2-hydroxyethyl)–1-piperazi-methanesulfonic acid, 2.5 MgATP, 0.3 NaGTP, 0.2 ethyleneglycoltetraacetic acid, 10 phosphocreatine. Access resistance ranged between 15 and 50 MΩ, and the results were discarded if the access resistance changed by >20%. Neurons recorded in hippocampal CA1 region were held at a holding potential of −70 mV. Series resistance (between 5–21 MΩ) was monitored throughout each experiment and neurons with a change in series resistance of more than 25% were excluded from the analysis.

To study intrinsic electrophysiological properties of CA1 inhibitory neurons, we analyzed the voltage responses to a series of hyperpolarizing and depolarizing square current pulses of 500ms duration of amplitudes between −70 pA and 130 pA with 10 pA step intervals from a holding potential of −70 mV in each cell. The voltage responses of neurons upon the current injections also helped to distinguish interneurons of a fast-spiking character having no or small 'sag' response to a hyperpolarizing current pulse from cells with lower spiking rate and a larger 'sag', typical of O-LM cells (*Maccaferri and McBain, 1996*; *Tables 1 and 2*). Input resistance (Rinput) was determined by calculating the slope of the plot of the membrane potential variation induced by a hyperpolarizing 500ms step of current (I: from −70 pA to 0 pA). Firing frequency was studied by injecting 500ms pulses of depolarizing current (I: from 10 pA up to 130 pA) into the cell and plotting the spike frequency (f) as a function of the current intensity (f/I plot). For presumed fast-spiking neurons, a second protocol was used were current steps varied from +100 pA to +400 pA with 100 pA steps, in order to analyze possible differences in high-speed discharge. Both analyses were done using Prism 8 software (GraphPad Software, La Jolla, CA). For the analysis of AP, the first AP evoked by a suprathreshold depolarizing current pulse (considered as the Rheobase) was selected. The following AP features were calculated using Minianalysis software: peak amplitude, area, half-width, threshold, AHP area, and AHP peak (*Tables 1 and 2*). We further analyzed the shape of the AP rising phase using multiple derivatives of the raw membrane potential signal using Clampfit software (PClamp, Molecular Devices). In some dV/dt curves appeared a shouldering. A 'double bump' in the d$^2$V/dt$^2$ curve and a value equal to or less than zero of the d$^3$V/dt$^3$ trace between the two bumps indicated that the shouldering corresponded to a stationary inflection in the rising slope of the AP. This indicated that somatic and axonal components of the AP could be temporally differentiated (*Kress et al., 2008*; *Paterno et al., 2021*).

GABA-induced spontaneous inhibitory post synaptic currents (sIPSCs) were recorded in pyramidal cells in voltage-clamp mode using patch pipettes of 8–10 MΩ filled with an internal solution containing the following (in mM): 140 CsCl, 1 MgCl2, 10 HEPES, 4 NaCl, 2 Mg-ATP, 0.3 Na-GTP, 0.1 EGTA. Access resistance ranged between 15 and 30 MΩ, and the results were discarded if the access resistance changed by >20%. The GABA-mediated current was pharmacologically isolated in the presence of AMPA and NMDA receptor antagonists (10 μM CNQX, 40 μM D-APV, respectively). For recording of miniature IPSCs (mIPSCs) TTX (1 μM) was added to the ACSF. CNQX, D-APV, and TTX were purchased from Tocris Bioscience (Bristol, UK).

## Statistical analysis

All values are given as means ± SEM. All statistical tests were performed using Prism 8 (GraphPad Software). For comparisons of two independent groups, we used Mann–Whitney test or Student's t test when values lie in a normal distribution using Shapiro-Wilk test. For multiple group comparisons, we used two-way ANOVA followed by the Sidak's multiple comparison test. To quantify the

probability that two sets of samples were drawn from the same probability distribution, we used the Kolmogorov–Smirnov test. To compare the rate of SST cells with stationary inflection to those without, we used Fisher's exact test.

## Acknowledgements

We are grateful to Dr Thomas Marissal for helpful discussions. We thank the University of California Davis/National Institutes of Health NeuroMab Facility and Developmental Studies Hybridoma Bank of the University of Iowa. This work was supported by the Fondation pour l'Aide à la Recherche sur la Sclérose en Plaques (ARSEP) to DP, CFS and DK.

## Additional information

### Competing interests

Giulia Bonetto: is affiliated with AstraZeneca. The author has no financial interests to declare. The other authors declare that no competing interests exist.

### Funding

| Funder | Grant reference number | Author |
|---|---|---|
| Fondation pour l'Aide à la Recherche sur la Sclérose en Plaques | Postdoc fellowship | Delphine Pinatel |
| Fondation pour l'Aide à la Recherche sur la Sclérose en Plaques | Grant | Domna Karagogeos Catherine Faivre-Sarrailh |

The funders had no role in study design, data collection and interpretation, or the decision to submit the work for publication.

### Author contributions

Delphine Pinatel, Catherine Faivre-Sarrailh, Conceptualization, Supervision, Funding acquisition, Investigation, Writing – review and editing; Edouard Pearlstein, Giulia Bonetto, Conceptualization, Investigation, Writing – review and editing; Laurence Goutebroze, Conceptualization, Resources, Investigation, Writing – review and editing; Domna Karagogeos, Valérie Crepel, Conceptualization, Resources, Writing – review and editing

### Author ORCIDs

Edouard Pearlstein http://orcid.org/0000-0001-9405-5667
Giulia Bonetto https://orcid.org/0000-0003-1469-2004
Valérie Crepel http://orcid.org/0000-0003-0408-3766
Catherine Faivre-Sarrailh http://orcid.org/0000-0002-1718-0533

### Ethics

The care and use of mice in all experiments were carried out according to the European and Institutional guidelines for the care and use of laboratory animals and approved by the local authority (laboratory's agreement number D13-055-8, Préfecture des Bouches du Rhône).

### Decision letter and Author response

Decision letter https://doi.org/10.7554/eLife.86469.sa1
Author response https://doi.org/10.7554/eLife.86469.sa2

## Additional files

### Supplementary files
• MDAR checklist

## Data availability

All data generated or analyzed during this study are included in the manuscript (source data files for Figure 1-9).

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
