## [Editor Report]

This important study identifies the functional consequence of myelination of interneuronal axons on circuit function by showing that 4.1B deletion leads to altered myelination in a subset of interneurons and altered intrinsic and synaptic physiological parameters. The authors' conclusions about how myelination of inhibitory axons affects physiological properties are based on solid evidence using a combination of imaging and electrophysiological approaches.

---

## [Decision Letter]

**Decision letter after peer review:**

Thank you for submitting your article "A class-specific effect of dysmyelination on the excitability of hippocampal interneurons" for consideration by *eLife*. Your article has been reviewed by 3 peer reviewers, and the evaluation has been overseen by a Reviewing Editor and Laura Colgin as the Senior Editor. The following individual involved in the review of your submission has agreed to reveal their identity: Maren Engelhardt (Reviewer #3).

Essential revisions:

1. The authors should refer to the interneuron population as SST-expressing neurons, or provide experimental evidence that they are indeed OLM cells.

2. Please, clarify issues related to the specificity of the immunostaining.

3. Please, clarify the disconnect between the measurements of the physical parameters and the details of the imaging parameters.

4. The authors should detail how specific these changes in myelinization are to local interneurons.

5. The behavioural data is only very weakly connected to the previous section of the study. From the current experimental design, it is unclear whether the observed changes are caused by alterations that occurred in the specific cell types and/or brain regions. For this reason, this section should be either strengthened with rescue experiments or removed from the manuscript.

*Reviewer #1 (Recommendations for the authors):*

1. The spiking of more PV interneurons should be recorded and compared. The number of PV interneurons may not be enough to assess the potential changes.

2. In the hippocampus, PV is expressed in axo-axonic cells in addition to basket cells and bistratified cells. As the authors have not identified the interneuron types expressing PV, it would be more appropriate to call these cells PV interneurons.

3. The investigations of sIPSCs and mIPSCs do not help strengthen the authors' claim. To test whether the OLM input onto pyramidal cells is indeed reduced, they may consider performing paired recordings from identified interneurons and pyramidal cells. A reduced number of sIPSCs with small amplitude is not a strong claim that OLM cell output is reduced in KO mice, since other interneurons also innervate distal dendrites of pyramidal cells. In addition, it is not clear what was the access resistance of neurons in this set of experiments. Please clarify.

4. I had difficulty understanding the staining shown in Figure 4E'. The image was taken from a section prepared from a 4.1B KO mice, yet this image shows a 4.1B immunostained profile pointed by an arrow. Either the 4.1B antibody is not specific, or this arrow points to a non-4.1B labelled profile. Please clarify.

5. Similarly, why do we see immunolabeling for Caspr in Caspr2-/- mice (Figure 6F)?

6. In many figures, there is a clear reduction in MBP labeling in the alveus/str. oriens in KO mice (e.g. Figure 3A, B), where the pyramidal axons run. Yet, it is claimed that there is no change in the number of stained profiles. Some results imply that pyramidal cells in the hippocampus express protein 4.1B, which would explain why there is a change in MBP staining in KO mice. Do the authors have a good method to exclude the possibility that MBP is not altered in pyramidal cell axons?

*Reviewer #2 (Recommendations for the authors):*

There are several pieces of data that the authors likely already have that would strengthen the claim about the specificity of the observed effects:

– In Figure 1 —figure supplement 1, panel D shows the percent of Lhx6 axons that are myelinated in the control and contactin2 KO mice. A similar figure with the percent of Lhx6 axons that are myelinated in the 4.1B KO mice will be very informative. If the % Lhx6 axons that are myelinated in the 4.1B KO mice is lower than in the control, that would argue for the specificity of the effect. Otherwise, if the myelin length is decreased, but the % myelinated Lhx6 axons is the same, that would be consistent with an overall decrease in axon myelination.

– More information on 4.1B distribution in the control hippocampus, if there is already data in the literature or any additional evidence that would point to a specific preference of 4.1B for PV and SST interneurons.

– Are there any changes at the AIS of PV neurons as the authors show for the SST neurons? Or in the AIS of pyramidal neurons (ankirynG stain)?

Increased amplitude of sIPSCs in the 4.1 KO – could also reflect an increased proximal inhibition, as mentioned in the results – but it is discussed only as evidence for the alternative, the decrease in distal inhibition.

An alternative interpretation of many of the observed effects is that 4.1B deficiency may be causing an overall delay in the development of myelination, as previously reported in the spinal cord. For example, the distribution of myelin in P25 hippocampus (Figure 1 —figure supplement 3) quite clearly shows that there is less myelin overall in the KO mouse. The oldest age used in the study is P70, and cortical myelination is known to continue beyond this age.

*Reviewer #3 (Recommendations for the authors):*

Specific suggestions for improving results and their presentation:

Two general comments:

1. The varying numbers of experimental animals (genotypes), ages, and experiments as well as parameters analyzed would benefit from a more structured presentation in the text, for example as a table. In its current form, the text requires a lot of back and forth to grasp the extent of data points presented in each figure.

2. For data presentation, I strongly suggest using data distribution plots/box plots in all figures. The authors have done so for Figures 2, 5, 7, 8, and 9, but nowhere else. Why?

In my opinion, the images provided in the manuscript are of very good anatomical value and prepared with precision and an eye for detail. My major point of conflict is the sampling of confocal images for the analysis of structural parameters. Nyquist conditions are essential to avoid undersampling in conventional confocal microscopy, especially when using the images to measure differences in the 1 μm range and smaller, in z. However, these conditions do not seem to be met with all data collected in this study, or I cannot find the required information. Using the stated information on p. 17, ll. 658 (40 x oil immersion objective, 1.32 NA and 488 nm excitation, assuming 1 as number of excitation photons and a lens immersion refraction index of 1.51), a quick calculation shows that single slices in a stack would require the z-stack sampling to be conducted at less than 0.2 μm. If this is not possible due to limitations of the confocal platform used, then single optical planes close to 0.3-0.5 should be produced, with subsequent deconvolution to overcome some of the limitations. However, the text states z-tack size of 2 μm (p. 17, ll. 669). This is problematic, especially since the parameters are used to quantify the density of such small structures as paranodes. I would like to encourage the authors to take a look at all their immunofluorescence data under these aspects and consider alternatives if indeed the Nyquist criterion is not met by a large factor.

Along the same lines, it is unclear in the current methods, how the 3D reconstruction was achieved prior to quantification. This pertains to myelin sheath length and AIS length. Particularly in the hippocampus, AIS are not present in seemingly organized orientation but are inherently "crooked" structures that cannot be measured (length, distance to soma, etc.) in merged intensity projections. Please state in more detail how length measurements were conducted (internodes, AIS). Again, for z-stack quantification, the Nyquist conditions become relevant. It is impossible to judge by stack size (as given for example on p. 18, ll. 682), where max intensity projections from 25 – 50 μm stacks are used to measure.

Regarding statistics, again, a table with summarized info on which experiment and quantification were done on how many animals, ROIs, or areas would be most helpful. Statistics are performed using either n for animals, or n for individual data points from several animals. Why is not all data represented as mean/animal? Also, the sampling in general with n = 3 animals is borderline acceptable; in some cases, it seems that only 2 animals were used (p. 18, ll. 687, 689), and in others, no number is given at all (p. 18, ll. 683 – n = 275 in control and n=191 in 4.1B-/- mice). This needs to be addressed, either by explaining why so few animals were used or by adding more data from individual animals. Assigning structures (AIS, nodes) as n results in overstating effects, since especially for AIS, there is significant heterogeneity in the length across neurons from the same type, and this is masked when 100 AIS are considered as individual n instead 100 AIS per animal, and the animal is (correctly) the n. Since the study seems to switch back and forth between these assignments, I suggest levelling these data across all experiments unless there are specific reasons not to do so, which then needs to be explained. As outlined on p. 20, all values are given as means {plus minus} SEM; this needs to be corrected for those cases where the standard deviation is the appropriate choice (e.g. all graphs showing n = individual structure, and not the mean of an animal). The data from electrophysiological recordings should be presented in such a way that e.g. the number of cells and/or animals is readily accessible from the graph or legend. In its current form, this information while available, needs to be pieced together from in-text information supplemented by figure legends. Sometimes, the authors do not include the number of animals behind individual cell data. For example, on p. 10, the paragraph beginning on l. 379 outlines data on inhibitory inputs on CA1 pyramidal neurons, which are statistically compared in the text (ll. 383), but do not highlight the number of cells or animals used. This information is also not available in the legend of Figure 9. Please carefully review all figures and edit accordingly.

---

## [Author Response]

Essential revisions:Reviewer #1 (Recommendations for the authors):1. The spiking of more PV interneurons should be recorded and compared. The number of PV interneurons may not be enough to assess the potential changes.

We conducted new patch-clamp experiments to include some additional fast spiking PV cells. It should be noted that due to the heterogeneity of PV cell discharge patterns within the stratum pyramidale (Tricoire et al., 2011), numerous recordings were required as a number of non-fast spiking cells were recorded that were not included in our analysis.

2. In the hippocampus, PV is expressed in axo-axonic cells in addition to basket cells and bistratified cells. As the authors have not identified the interneuron types expressing PV, it would be more appropriate to call these cells PV interneurons.

We changed PV basket cells into PV cells throughout the text (Summary, lane 62, Introduction, lane 131, Results, lane 342, Discussion, lanes 547, 549).

We now mention the axo-axonic cells in the Results, lane 329:

“Different subtypes of Lhx6interneurons are located in the stratum pyramidale, including the fast-spiking basket cells innervating the soma of pyramidal cells, axo-axonic cells contacting their AIS, and bistratified cells contacting their basal and apical dendrites.”

3. The investigations of sIPSCs and mIPSCs do not help strengthen the authors' claim. To test whether the OLM input onto pyramidal cells is indeed reduced, they may consider performing paired recordings from identified interneurons and pyramidal cells. A reduced number of sIPSCs with small amplitude is not a strong claim that OLM cell output is reduced in KO mice, since other interneurons also innervate distal dendrites of pyramidal cells. In addition, it is not clear what was the access resistance of neurons in this set of experiments. Please clarify.

We agree with the referee that the OLM neurons are not the sole one connecting PC distal dendrites since some CCK interneurons were also reported to project to the distal dendrites (Somogyi and Klausberger, 2005). This comment was added in the Discussion, lane 572:

“Functionally, it is well established that PV interneurons are involved in proximal inhibition while SST and also some CCK interneurons take part in distal dendritic inhibition”.

Regarding the access resistance, the recording conditions were now indicated in the Methods:

“…values ranged between 15 and 30 MΩ, and the results were discarded if the access resistance changed by >20%.” (Methods, lane 783)

4. I had difficulty understanding the staining shown in Figure 4E'. The image was taken from a section prepared from a 4.1B KO mice, yet this image shows a 4.1B immunostained profile pointed by an arrow. Either the 4.1B antibody is not specific, or this arrow points to a non-4.1B labelled profile. Please clarify.

Sorry for the misleading legend, Figure 4E and E’ are WT. We modified the panel to make it clear and added an image of immunostaining for 4.1B along SST axons (new Figure 4F).

5. Similarly, why do we see immunolabeling for Caspr in Caspr2-/- mice (Figure 6F)?

Caspr and Caspr2 are encoded by two different genes. Deletion of Caspr2 does not induce paranodal alteration, Caspr is expressed at paranodes in the Caspr2-/- mice as in WT.

6. In many figures, there is a clear reduction in MBP labeling in the alveus/str. oriens in KO mice (e.g. Figure 3A,B), where the pyramidal axons run. Yet, it is claimed that there is no change in the number of stained profiles. Some results imply that pyramidal cells in the hippocampus express protein 4.1B, which would explain why there is a change in MBP staining in KO mice. Do the authors have a good method to exclude the possibility that MBP is not altered in pyramidal cell axons?

We have quantified the paranodal density in the alveus as shown in the new Figure 1I. See Results lane 178:

“The density of paranodes was not affected in the alveus (P=0.8002) nor in the stratum lacunosum-moleculare (P=0.6273) suggesting that myelinated axons connecting extra-hippocampal areas may be preserved. In particular, this is an indication that the axons of pyramidal neurons that run into the alveus should be properly myelinated.”

We also quantified the length of myelinated axons in the stratum oriens as shown in the new Figure 1J.

“Myelination was significantly decreased by 47% (P=0.0018, Student’s t test) in the stratum oriens” (See Results lane 185).

Reviewer #2 (Recommendations for the authors):There are several pieces of data that the authors likely already have that would strengthen the claim about the specificity of the observed effects:– In Figure 1 —figure supplement 1, panel D shows the percent of Lhx6 axons that are myelinated in the control and contactin2 KO mice. A similar figure with the percent of Lhx6 axons that are myelinated in the 4.1B KO mice will be very informative. If the % Lhx6 axons that are myelinated in the 4.1B KO mice is lower than in the control, that would argue for the specificity of the effect. Otherwise, if the myelin length is decreased, but the % myelinated Lhx6 axons is the same, that would be consistent with an overall decrease in axon myelination.

We evaluated the percentage of Lhx6-positive myelinated axons in the stratum radiatum of 4.1B KO mice (Figure 1—figure supplement 1D), which appeared nearly similar to control (66.1 ± 3.6% in 4.1B KO versus 75 ± 5% in control) as an indication that myelin alteration was not restricted to Lhx6 GABAergic axons. Results lane 189.

– More information on 4.1B distribution in the control hippocampus, if there is already data in the literature or any additional evidence that would point to a specific preference of 4.1B for PV and SST interneurons.

To our knowledge, this is the first study that examines the expression and functional role of 4.1B in GABAergic neurons in vivo. We previously described the enrichment of 4.1B associated with Kv1 channels in PV and SST inhibitory axons of hippocampal cell culture (Bonetto et al. 2019).

– Are there any changes at the AIS of PV neurons as the authors show for the SST neurons? Or in the AIS of pyramidal neurons (ankyrinG stain)?

The AIS of SST neurons can be easily detected isolated in the stratum oriens. In contrast, it is very difficult to image the AIS of PV cells, which are located within the pyramidal layer, intermingled with the numerous AIS of pyramidal cells.

Increased amplitude of sIPSCs in the 4.1 KO – could also reflect an increased proximal inhibition, as mentioned in the results – but it is discussed only as evidence for the alternative, the decrease in distal inhibition.

This sentence was modified in the Discussion, lane 576:

“This suggests that either the distal inhibitory inputs onto pyramidal cells may be decreased as a consequence of the reduced excitability of SST cells or that the proximal inhibition by PV cells may be increased. Nevertheless, this last hypothesis is unlikely since we did not observe an increase in the excitability of fast-spiking PV interneurons.”

An alternative interpretation of many of the observed effects is that 4.1B deficiency may be causing an overall delay in the development of myelination, as previously reported in the spinal cord. For example, the distribution of myelin in P25 hippocampus (Figure 1 —figure supplement 3) quite clearly shows that there is less myelin overall in the KO mouse. The oldest age used in the study is P70, and cortical myelination is known to continue beyond this age.

We have now examined the myelin pattern of 4.1B KO mice at an older age. Results lane 171: Such myelin pattern alteration persisted in mature animals as shown at the age of 6 months (Figure1—figure supplement 4).

Reviewer #3 (Recommendations for the authors):Specific suggestions for improving results and their presentation:Two general comments:1. The varying numbers of experimental animals (genotypes), ages, and experiments as well as parameters analyzed would benefit from a more structured presentation in the text, for example as a table. In its current form, the text requires a lot of back and forth to grasp the extent of data points presented in each figure.2. For data presentation, I strongly suggest using data distribution plots/box plots in all figures. The authors have done so for Figures 2, 5, 7, 8, and 9, but nowhere else. Why?

For clarity of data presentation, bars are shown in Figure 1 (n = 4 mice) and Figure 3. Data are shown as box plots in Figure 6 (n = 7-11 ROIs).

In my opinion, the images provided in the manuscript are of very good anatomical value and prepared with precision and an eye for detail. My major point of conflict is the sampling of confocal images for the analysis of structural parameters. Nyquist conditions are essential to avoid undersampling in conventional confocal microscopy, especially when using the images to measure differences in the 1 μm range and smaller, in z. However, these conditions do not seem to be met with all data collected in this study, or I cannot find the required information. Using the stated information on p. 17, ll. 658 (40 x oil immersion objective, 1.32 NA and 488 nm excitation, assuming 1 as number of excitation photons and a lens immersion refraction index of 1.51), a quick calculation shows that single slices in a stack would require the z-stack sampling to be conducted at less than 0.2 μm. If this is not possible due to limitations of the confocal platform used, then single optical planes close to 0.3-0.5 should be produced, with subsequent deconvolution to overcome some of the limitations. However, the text states z-tack size of 2 μm (p. 17, ll. 669). This is problematic, especially since the parameters are used to quantify the density of such small structures as paranodes. I would like to encourage the authors to take a look at all their immunofluorescence data under these aspects and consider alternatives if indeed the Nyquist criterion is not met by a large factor.Along the same lines, it is unclear in the current methods, how the 3D reconstruction was achieved prior to quantification. This pertains to myelin sheath length and AIS length. Particularly in the hippocampus, AIS are not present in seemingly organized orientation but are inherently "crooked" structures that cannot be measured (length, distance to soma, etc.) in merged intensity projections. Please state in more detail how length measurements were conducted (internodes, AIS). Again, for z-stack quantification, the Nyquist conditions become relevant. It is impossible to judge by stack size (as given for example on p. 18, ll. 682), where max intensity projections from 25 – 50 μm stacks are used to measure.

Concerning the measurement of internodes, large stacks (25-50 µm width) have been imaged using the 20x objective. Paranodes were visible on two consecutive confocal sections using 2-µm steps. Only the internodes bordered by paranodes at each end were selected and measured on 2D maximum projection images.

Concerning the measurement of AIS, we hope that we have now better illustrated the AIS of SST cells in the stratum oriens in the new Figure 8 showing single channel images. In contrast to the AIS of pyramidal neurons which display sinuous feature, the AIS of SST neurons (and especially O-LM cells which axons run straight across the stratum radiatum) show a rather straight organization.

We improved the measurements of the AIS structural parameters (onset, length) of SST neurons of the stratum oriens and extended the Method section:

“To encompass AIS onset and distal sites, we selected neurons with SST axonal staining exceeding AnkyrinG AIS staining. The bottom and top sections were carefully determined for z-stack acquisition (20x objective, 0.54 µm steps, Nyquist conditions). The AIS length was measured on 2D projection taking care that the AIS of SST neurons were orientated in the plane of section (z-width 2-8 µm; means ± SD: WT, 4.74±2.23; 4.1B KO, 4.58±1.56). Measurements were performed on stacks using the ImageJ “synchronize windows” tool.” Methods, lane 723.

Indeed, these new measurements confirm that the AIS of SST neurons was significantly shorter in the 4.1B KO mice.

Regarding statistics, again, a table with summarized info on which experiment and quantification were done on how many animals, ROIs, or areas would be most helpful. Statistics are performed using either n for animals, or n for individual data points from several animals. Why is not all data represented as mean/animal? Also, the sampling in general with n = 3 animals is borderline acceptable; in some cases, it seems that only 2 animals were used (p. 18, ll. 687, 689), and in others, no number is given at all (p. 18, ll. 683 – n = 275 in control and n=191 in 4.1B-/- mice). This needs to be addressed, either by explaining why so few animals were used or by adding more data from individual animals. Assigning structures (AIS, nodes) as n results in overstating effects, since especially for AIS, there is significant heterogeneity in the length across neurons from the same type, and this is masked when 100 AIS are considered as individual n instead 100 AIS per animal, and the animal is (correctly) the n. Since the study seems to switch back and forth between these assignments, I suggest levelling these data across all experiments unless there are specific reasons not to do so, which then needs to be explained. As outlined on p. 20, all values are given as means {plus minus} SEM; this needs to be corrected for those cases where the standard deviation is the appropriate choice (e.g. all graphs showing n = individual structure, and not the mean of an animal). The data from electrophysiological recordings should be presented in such a way that e.g. the number of cells and/or animals is readily accessible from the graph or legend. In its current form, this information while available, needs to be pieced together from in-text information supplemented by figure legends. Sometimes, the authors do not include the number of animals behind individual cell data. For example, on p. 10, the paragraph beginning on l. 379 outlines data on inhibitory inputs on CA1 pyramidal neurons, which are statistically compared in the text (ll. 383), but do not highlight the number of cells or animals used. This information is also not available in the legend of Figure 9. Please carefully review all figures and edit accordingly.

For the measurement of internode length, n = 275 in control and n=191 in 4.1B-/-; means ± SEM of 4 ROIs; 2 mice/genotype as indicated in the Methods and in the Legend of Figure 4.

For the electrophysiological recordings, the number of cells is now indicated in the text, and mice and cell number are indicated in Table 1 and Table 2 as noted in the Legend of Figure 7:

“For a detailed statistical summary of intrinsic parameters, see Table 1 and Table 2. (n=6-9 mice/genotype)”.

For Figure 8, (n=12 cells from 9 mice and n=17 cells from 8 4.1B^-/-^ mice) was added in the Legend. For Figure 9: 6 cells and 4-5 mice/genotype was indicated.

For the quantitative morphological analysis: We considered “n = number of animals” in Figure 1 to describe the massive and selective alteration of myelin in the hippocampus of 4.1B KO mice. This phenotype is marked and highly penetrant in all the mutant mice examined. In the following Figures, we choose to consider “n=ROIs” (Figure 2, Figure 3, Figure 6) for the density of SST and PV interneurons or oligodendroglial cells and “n = individual structures” (Figure 4, Figure 5, Figure 8) for a more precise sampling of the structure heterogeneity (internode, node, AIS). Means ± SEM are indicated in the text corresponding to plot boxes and distribution plots in the Figures.